# Urinary metabolite model to predict the dying process in lung cancer patients
Séamus Coyle [1,2] ✉, Elinor Chapman [3], David M. Hughes [4], James Baker [3], Rachael Slater [3], Andrew S. Davison [5,6], Brendan P. Norman [6], Ivayla Roberts [3,7], Amara C. Nwosu [8,9,10], James A. Gallagher[6], Lakshminarayan R. Ranganath [5,7], Mark T. Boyd [1,3], Catriona R. Mayland [10,11], Douglas B. Kell [7,12], Stephen Mason [10], John Ellershaw [10,13] & Chris Probert [3]

## Abstract

**Background** Accurately recognizing that a person may be dying is central to improving their experience of care at the end-of-life. However, predicting dying is frequently inaccurate and often occurs only hours or a few days before death.

**Methods** We performed urinary metabolomics analysis on patients with lung cancer to create a metabolite model to predict dying over the last 30 days of life.

**Results** Here we show a model, using only 7 metabolites, has excellent accuracy in the Training cohort $n = 112$ (AUC = 0·85, 0·85, 0·88 and 0·86 on days 5, 10, 20 and 30) and Validation cohort $n = 49$ (AUC = 0·86, 0·83, 0·90, 0·86 on days 5, 10, 20 and 30). These results are more accurate than existing validated prognostic tools, and uniquely give accurate predictions over a range of time points in the last 30 days of life. Additionally, we present changes in 125 metabolites during the final four weeks of life, with the majority exhibiting statistically significant changes within the last week before death.

**Conclusions** These metabolites identified offer insights into previously undocumented pathways involved in or affected by the dying process. They not only imply cancer's influence on the body but also illustrate the dying process. Given the similar dying trajectory observed in individuals with cancer, our findings likely apply to other cancer types. Prognostic tests, based on the metabolites we identified, could aid clinicians in the early recognition of people who may be dying and thereby influence clinical practice and improve the care of dying patients.

## Plain Language Summary

Recognizing when someone is nearing the end of life is important for providing better care, but it is often hard to predict accurately. In our study, we analyzed urine samples from lung cancer patients to develop a method that can predict when a person is in their last 30 days of life. We identified 7 key chemicals in the urine that helped us predict death with high accuracy. This method worked better than current tools and provided reliable predictions throughout the last month of life. We also found changes in many other chemicals in the final weeks. These findings could help doctors identify when a patient is dying earlier, leading to better care at the end of life.

Accurately recognizing that a person may be dying and in the last weeks or days of life is central to improving peoples' experience of care. It enables families, medical teams and health-care providers to plan and provide the best care possible. However, physicians' predictions of dying are frequently inaccurate and overoptimistic[1]. The 2019 United Kingdom National Audit of Care at the End of Life in hospitals found the recognition of the dying was

challenging. When dying was recognized, 20% of people died within 8 h; the median time to death was 36 h; and importantly 50% of patients lacked the capacity to be directly involved in any decision-making[2].

Predicting when a patient with advanced cancer is likely to die is a challenge and currently no diagnostic test is available. Globally, there were 19·3 million new cancer cases and almost 10 million cancer deaths in 2020;

[1]Liverpool Head and Neck Cancer Centre, University of Liverpool, Liverpool, UK. [2]Department of Palliative Medicine, Clatterbridge Cancer Centre, Liverpool, UK. [3]Institute of Systems, Molecular and Integrative Biology, University of Liverpool, Liverpool, UK. [4]Department of Health Data Science, University of Liverpool, Liverpool, UK. [5]Department of Clinical Biochemistry and Metabolic Medicine, Liverpool Clinical Laboratories, Liverpool University Hospitals Foundation Trust, Liverpool, UK. [6]Institute of Life Course & Medical Sciences, University of Liverpool, Liverpool, UK. [7]Centre for Metabolomics Research (CMR), Department of Biochemistry, Cell and Systems Biology, Institute of Systems, Molecular and Integrative Biology, University of Liverpool, Liverpool, UK. [8]Lancaster Medical School, Lancaster University, Lancaster, UK. [9]Marie Curie Hospice, Liverpool, UK. [10]Palliative Care Unit, Institute of Life Course & Medical Sciences, University of Liverpool, Liverpool, UK. [11]Division of Clinical Medicine, School of Medicine & Population Health, University of Sheffield, Sheffield, UK. [12]Novo Nordisk Foundation Centre for Biosustainability, Danish Technical University of Denmark, Lyngby, Denmark. [13]Academic Palliative & End of Life Care Department, Liverpool University Hospitals NHS Foundation Trust, Liverpool, UK. ✉e-mail: s.coyle@liverpool.ac.uk

lung cancer had the highest mortality, responsible for 1·8 million deaths[3]. Accurate prognostic information at the end of life is essential to co-ordinate and manage care in response to need, whilst avoiding burdensome and unnecessary interventions. Several validated prognostic tools aim to predict the survival of patients with advanced cancer[4]. A recent comparison of five validated prognostic tools showed the best model, PiPs-B (based on clinical observations and blood results), was as accurate as expert multidisciplinary clinician judgement[5]. However, existing models only consider a binary outcome of death from a particular time point e.g. 30 days. An objective model estimating risk of death over a range of time periods including the last days of life is needed.

We do not know how people die from cancer. In the last 2 weeks of life, there is evidence for deranged respiratory and renal function variables[6], although few patients have evidence of organ failure. Pulmonary embolus and infection are thought to be the major causes of death based on post-mortem studies[7,8]. However, it is unusual for people with cancer to die suddenly as anticipated from a pulmonary embolus. About a third of patients with advanced cancer admitted to specialist palliative care units have a femoral deep vein thrombosis. Therefore, thromboembolism is considered a manifestation of advanced disease, rather than a cause of premature death[9]. There is also a difference between the physiological deterioration leading to death in the acutely unwell patient compared to people dying from cancer; there is no evidence of sepsis in those with cancer[6]. This suggests cancer patients generally do not die from organ failure and in those that die with an infection or pulmonary embolus die with them, rather than from these events.

A systematic review of biomarkers associated with dying identified common themes in cancer patients, irrespective of the type of malignancy: systemic inflammation, haematological changes, cachexia, hepatic dysfunction, renal dysfunction, and electrolyte changes[10]. Given the common features shared in patients dying from cancer, a "dying process" has been proposed[10] but the biochemical pathways involved have not yet been described.

We hypothesized a dying process is associated with metabolic changes. Here, we developed a model to predict dying based on urinary metabolites: training and validation data are presented. Some of the metabolites are known to be involved in particular pathways and the relevant pathways are described and discussed.

## Materials and methods
### Study design
Urine samples from patients with lung cancer towards the end of life were prospectively collected. Untargeted urine metabolomic studies were conducted using liquid chromatography quadrupole time of flight mass spectrometry (LC-QTOF-MS) to identify metabolites in the urine. To identify the metabolites that change in the last weeks of life, we applied two approaches; ANOVA and volcano plot analysis. To predict the last days of life, a Cox proportional hazards model with a lasso penalty was developed and used. The prediction model was validated on a separate cohort that was collected and analyzed after the initial ("Training") cohort. This validation cohort was also analyzed in a different laboratory using an Orbitrap mass spectrometer. Further KEGG Pathway Analysis was performed, using the statistically significant metabolites identified for the last 2 weeks and last 3 days of life. To identify the pathways affected during the dying process, we then summarized and collated findings from the Volcano plot analysis, ANOVA analyses and KEGG pathway analysis to identify pathways affected during the dying process.

### Setting and participants
The study was conducted at eight sites (hospitals and hospices) in the North West of England (UK) from June 1st 2016 to March 31st 2020. Ethical approval was provided by North Wales (West) Research Ethics Committee (REC reference 15/WA/0464). Patients with incurable lung cancer (locally or metastatic) were recruited prospectively with their informed consent and urine samples were collected up to three times a week[11]. We approached

every adult with advanced lung cancer who the treating team thought would not be distressed discussing the study, could understand and communicate in English and had capacity to provide informed consent. Written informed consent was obtained. Urine samples were not collected at set times during the day and it was not a 24 h collection. Patients who were often frail, could not always supply a sample at a set time. The list of medications a patient was on for the previous 24 h was recorded for each sample. Each sample was retrospectively assigned a day before death when this was known. In order not to bias the results, only one sample per patient, the one closest to death, was included for analysis. In metabolomics studies, analyzing only one sample per subject, especially in large datasets with variable numbers of samples from different subjects like ours, is aimed at ensuring that the data accurately reflects the biological variation of interest, while minimizing biases that can arise from unequal sampling and non-independence of observations. The Validation cohort was recruited after the initial Training cohort and analyzed over 2 years after the Training cohort.

### Urine collection, storage and preparation
Random urine samples were collected twice per week from participants into universal (20 mL) and glass (5 mL) containers. An anonymized record of medication administered in the previous 24 h was recorded. Samples were stored on site at -20 °C before subsequently being transferred to the University of Liverpool for further storage at -80 °C. Individual patient samples were thawed at room temperature, vortexed and supernatants separated into four replicate aliquots in individual 96-well plates (Waters, UK) which were stored at -80 °C until analysis by one of four different methods; two different chromatography conditions in negative and positive ionization polarity. Pooled quality control samples were created following the protocol described by Norman et al.[12]. For each sample group (time before death), a separate representative pool was created by pooling an equal volume of each individual urine sample for quality control purposes. An overall pool was also created by pooling equal proportions of the above group pools. Analysis of individual and pooled samples was performed following dilution of 1:3 urine:deionised water (DIRECT-Q 3UV Millipore water purification system)[12].

### LC-QTOF-MS method
Untargeted urine metabolomic studies were performed using liquid chromatography quadrupole time of flight mass spectrometry (LC-QTOF-MS)[12]. In brief, analysis was performed on an Agilent 1290 Infinity LC coupled to an Agilent 6550 QTOF-MS equipped with a dual AJS electrospray ionization source (Agilent, UK).

**Liquid chromatography conditions.** Liquid chromatography (LC) methods were run in positive and negative polarity. LC method 1: employed an Atlantis dC18 column (3 × 100 mm, 3 μm, Waters, UK) maintained at 60 °C with flow rate at 0.4 mL/min. Mobile phases were (A) water and (B) methanol both containing 5 mmol/L ammonium formate and 0·1 % formic acid. The elution gradient started at 5 % B at 0 to 1 min increasing linearly to 100 % by 12 min, held at 100 % B until 14 min, returning to 95 % A for 5 min. LC method 2: used a BEH amide column (3 × 150 mm, 1·7 μm, Waters, UK) maintained at 40 °C with flow rate at 0·6 mL/min. Mobile phases were (A) water and (B) acetonitrile both containing 0.1 % formic acid. The elution gradient started at 99 % B, decreasing linearly to 30 % from 1 – 12 min, held at 30 % B until 12·6 min, returning to 99 % B for 3·4 min. Sample injection volume was 1 μL for both LC methods. The sampling needle was washed with a solution of water:methanol:isopropanol (45:45:10 v/v) between injections.

**Mass spectrometry conditions.** The mass spectrometer was tuned and calibrated according to protocols recommended by the manufacturer. Acquisition was performed in 2 *GHz* mode and mass range 50–1700. The capillary voltage was 4000 V and fragmentor voltage 380 V. The desolvation gas temperature was 200 °C with flow rate at 15 L/min. The sheath gas temperature was 300 °C with flow rate at 12 L/min. The nebulizer

pressure was 40 psig and nozzle voltage 1000 V (± for positive and negative ionization modes). The acquisition rate was 3 spectra/second. The reference mass solution was continually infused at a flow rate of 0·5 mL/min by a separate isocratic pump for constant mass correction. Each analytical run commenced with 20 injections of the overall pooled sample for system equilibration. The order of individual samples was randomized computationally and pooled samples were interspersed throughout the analytical sequence, every 10th injection. Injections of each sample group pool and the overall pooled sample were also placed at the start (post-equilibration) and end of each analytical sequence[12].

**Data pre-processing and quality control.** All data were acquired using the MassHunter suite (Agilent build 6.0) with quality checks being performed by Qualitative Analysis (build 07.00). Mass accuracy was checked using extracted ion chromatograms of reference masses: the resulting accuracy was ±5 ppm during the run. Additionally, chromatographic reproducibility was checked by overlaying binary pump pressure curves across each analytical sequence. Data was filtered based upon the pooled QC samples, with compounds being retained if observed in 100 % of replicate injections for at least one pool and with peak area coefficient of variation (CV) < 25 % across all replicate injections for each pool[13].

**Feature extraction.** Targeted feature extraction was performed on each dataset based on matching of metabolite chemical features against an in-house compound library accurate mass and retention time (AMRT) database that included a broad range of metabolites involved in intermediary metabolism. This was previously generated from analysis of the IROA Technology MS compound library of standards by each LC method described above, combined with the same QTOF analytical parameters used in this study[12] (databases publicly available: https://doi.org/10.6084/m9.figshare.c.4378235.v2). In addition to accurate mass and retention time, tandem mass spectrometry (MS/MS) was also used in the confirmation of metabolite identity (i.e. level 1 identification) as per Sumner et al.[14]. Compound identification of unknown metabolites in our model was attempted by matching of our acquired MS/MS spectra against the spectral libraries METLIN[15] (Agilent METLIN metabolite PCDL accurate mass library (build 07.00)), METLIN online[16], MoNA[17] and also using SIRIUS software (version 5.6.3) tools including the HMDB database, SIRIUS molecular formula identification and CSI:FingerID fingerprint prediction.

Data-dependent tandem mass spectrometry (MS/MS) was performed for metabolite identification by analysis of the pooled samples with highest abundance of compounds of interest. Accurate mass precursor ion targets were [M + H]+ and [M − H]−. Multiple fixed collision energies were applied (10–40 eV) with acquisition rates 6 spectra/second in MS1 and 4 spectra/second in MS/MS.

**Cross-laboratory validation**
**Sample preparation.** Urine samples were extracted with methanol protein crush by adding 75 μL of water and 75 μL of methanol to 50 μL of urine. Diluted samples were centrifuged at 4500 rpm for 20 min at 4 °C and 150 μL of supernatant was transferred in 96-well plate for analysis.

**LC Orbitrap mass spectrometer data acquisition.** Untargeted LC Orbitrap MS data was acquired using ThermoFisher Scientific Vanquish UHPLC system coupled to ThermoFisher Scientific ID-X Tribrid mass spectrometer (ThermoFisher Scientific, UK) following published guidelines[13,18–22]. LC method used the same column, conditions and solvents as described in the previous section. The LC gradient was also replicated to match the initial study. In difference to the original method and in adherence to laboratory guidelines the sample injection volume was set to 2 μL and the needle wash solution was composed of water:-methanol:isopropanol:acetonitrile (1:1:1:1 v/v).

Full-scan MS data was acquired in the Orbitrap mass analyser in the m/z range 66.7–1,000 with a mass resolution of 120,000 Full Width Half

Maximum (FWHM) at m/z = 200, a chromatographic peak width of 4 s (FWHM), Normalized AGC target (%) = 50 and maximum injection time = 100 ms. Source and ion transfer parameters applied were as follows: Sheath flow rate (arbitrary units) = 40, auxiliary gas flow rate (arbitrary units) = 8, sweep gas flow rate (arbitrary units) = 1, spray voltage = 3.5 kV (positive) and -3.0 kV (negative), capillary temperature (°C) = 275, S-lens RF level (%) = 45, auxiliary gas heater temperature (°C) 320, source position = M2.

Data-dependent MS/MS data acquisition was performed on group pooled samples at the end of the analytical batch. Data was acquired in the Orbitrap mass analyser with a mass resolution of 60,000 Full Width Half Maximum (FWHM) at m/z = 200 for the master scan and 30,000 FWHM at m/z 200 for MS/MS, both with a chromatographic peak width to 4 s (FWHM), Normalized AGC target (%) = 50, maximum injection time = 54 ms, cycle time (sec) = 0.6, isolation window = 1.5 m/z, stepped HCD = 20, 40 and 60, AGC target Standard, intensity threshold = $2 \times 10^4$, exclude isotopes = on and dynamic exclusion = 6.0 s. Additionally, the method included Targeted Mass and Targeted Mass Exclusion steps with 10 ppm tolerance for both low and high limits. For both steps the AcquireX option was turned on.

The ThermoFisher Scientific AcquireX deep scan acquisition workflow was employed on samples pooled by analysed group i.e., patients in the last month before death and patients within 3 months of death. AcquireX parameters were kept to default values except for Inclusion List Peak Window Extension which was set to 10 s and Exclusion Duration was also set to 10 s. AcquireX workflow enables a more complete acquisition of unique features in samples by using automated and iterative data dependent MS/MS acquisition. This is achieved by using automatically updated run-to-run inclusion and exclusion lists. In this study the use of AcquireX allowed us to acquire an MS2 spectra for 95% and 70% of the retained features in ESI + and ESI- respectively.

**Data processing and quality control.** Raw instrument data in.RAW file format were exported to ThermoFisher Scientific Compound Discoverer 3.3 (CD3.3) for deconvolution, alignment and annotation[23]. Compound identification was performed against ThermoFisher Scientific mzCloud spectral library with a score >70% (MSI 2) or against in-house spectral library with a score >75% (MSI 1) and full match on the proposed molecular formula from CD 3.3. For all data acquired, annotation and identification criteria were according to Sumner et al.[14]. Compounds used in the predictive model were matched based on identification and when not available as in the case of unknown metabolites, m/z, RT and MS/MS similarity were used.

**Statistics and reproducibility**
The sample size was initially determined by a feasibility study. The sample size required was calculated using a multivariate statistical method called probabilistic principal component analysis via the R package MetSizeR, designed specifically for metabolomics experiments based on nuclear magnetic resonance and mass spectrometry. For CoxLASSO modelling, missing values were replaced by one-half of the minimum positive value for each variable, data were normalized by probabilistic quotient normalization (PQN), autoscaled and glog transformed. Volcano and ANOVA analysis were performed using MetaboAnalyst[24], missing values were replaced by one-fifth of the minimum positive value for each variable. Data were normalized by median log10 transformed and auto-scaled. Unequal variance was assumed. Volcano plots measured differences in compound abundance: fold change (FC > 2, FC < −2) and $p < 0.05$ (false discovery rate [FDR] adjusted) were considered statistically significant. Analysis of Variance (ANOVA) investigated differences in metabolite abundance in the last weeks of life compared to samples >12 weeks from death. Graphs were generated using ggplot on R Studio version 1.4.1717[25]. Although multiple samples were collected from patients, only the final sample was included in the analysis to satisfy the modelling assumption of independence of observations.

## Prediction modelling

PROBAST guidelines for reporting prediction models were followed[26]. A Cox proportional hazards model with lasso penalty to derive a prediction model was used for assessing the last days of life in our cohort[27]. This approach is similar to the standard Cox model but shrinks parameter estimates towards zero, reducing over-fitting due to the large number of potential metabolites to consider as possible predictors of death. Administrative censoring was applied if the individual was still alive 30 days after their sample was supplied.

A penalty parameter (lambda) was imposed to determine the amount of smoothing chosen when 10-fold cross-validation was performed. The value of lambda that gave minimum mean cross-validated error was used for both the prediction model and internal validation.

The model was internally validated using bootstrap resampling methods with 1000 bootstrap samples. The penalty parameter was fixed from the original Cox lasso model to fit to the whole dataset, and then, for each bootstrap sample, a Cox lasso model was fitted and time-dependent area under the curve was calculated. Model calibration was assessed with each bootstrap sample by comparing the observed and expected survival probabilities, splitting the predicted risks into 3 groups (denoted low, medium and high risk of dying). The Validation cohort was externally validated. Calibration was performed at 10, 20 and 30 days. Kaplan–Meier curves were used to visually inspect the survival probabilities based on 30 day predicted risk. Log-rank tests were used to statistically compare the survival curves. Analysis was performed in R Studio version 1.4.1717 and used the packages "glmnet", "survival", and "hdnom"[25].

## Pathway analysis

The pathway analysis was undertaken using MetaboAnalyst version 5·0[24] and combines pathway enrichment analysis with pathway topology analysis to identify the most relevant pathways involved with the conditions under study. Metabolites that showed a statistically significant difference between groups were collated into relevant human KEGG physiological pathways to visualise which were altered towards dying. Metabolites were matched using the publicly available HMDB, PUB-CHEM and KEGG databases.

## Reporting summary

Further information on research design is available in the Nature Portfolio Reporting Summary linked to this article.

# Results
## Patients

We prospectively collected multiple samples of urine from patients with lung cancer. To decrease confounding factors this study only recruited lung cancer patients. TNM staging data were not collected in this study as these are used at cancer diagnosis to predict 5 year survival and guide clinician treatment. We aimed to collect samples from inpatients 2–3 times a week. This was possible with a small number of patients. For the majority of patients, the collection of repeated samples in the last weeks of life was incredibly difficult to achieve. We were dependent on nurses and health care assistants to remember in their busy schedules to collect a sample or for research nurses to be on site—which may have only been twice a week for half a day. Patients towards the end of life were very frail, for some they may have only urinated once or twice a day and could not supply a sample at a set time. In addition, some patients when they deteriorated were unable to supply a sample unless they were catheterised. This all meant that despite the intention to collect repeated samples over time, this was not always possible for the majority of patients. Only 3 patient samples analysed were on some form of chemotherapy (afatinib, pembrolizumab and exemethasane) and none of these samples were in the last 6 weeks of life. There were two independent cohorts; 112 patients in the Training cohort and 49 patients in the Validation cohort (Table 1). Our analysis was in four parts, (i) reporting metabolites that change in the last weeks of life, (ii) developing a training model to predict death, (iii) validating the model in a separate cohort of

patients and (iv) identifying biochemical pathways affected in the last weeks of life.

## Metabolites that change in the last weeks

Our first aim was to investigate whether metabolites change in the last weeks of life. Using the Training cohort ($n = 112$) we applied two approaches; ANOVA and volcano plot analysis. ANOVA is beneficial for comparing means across multiple groups and understanding sources of variability, while volcano plot analysis is valuable for visualizing and prioritizing features in high throughput datasets. ANOVA analysis identified 93 metabolites that varied in abundance. The abundance of 87 (94%) metabolites differed significantly in the last week of life (week 1) compared to week 2, week 3, week 4, week 4+ (weeks 4–11) or week 12+ (see summary in Supplementary Data File 1 and box-plots in Supplementary Figs. 1–10 in the Supplementary Information). Volcano plot analysis identified 85

**Table 1 | Clinical characteristics of the patients in the training cohort and the validation cohort included in this study**

| Patients | Training cohort Absolute number / 112 (%) | Validation cohort Absolute number / 49 (%) |
|---|---|---|
| **Sex** | | |
| Female: Male | 45 (40): 67 (60) | 27 (55): 22 (45) |
| **Diagnosis** | | |
| NSCLC[a] (Adenocarcinoma) | 35 (31) | 2 (4) |
| NSCLC[a] (Squamous) | 27 (24) | 1 (2) |
| NSCLC[a] (Unspecified) | - | 18 (37) |
| SCLC[b] | 22 (20) | 11 (22) |
| Mesothelioma | 3 (3) | 2 (4) |
| Radiological Diagnosis[c] | 25 (22) | 15 (31) |
| **Age (years)** | | |
| Median (range) | 71 (47–89) | 71 (48–94) |
| 40–49 | 4 (4) | 1 (2) |
| 50–59 | 14 (13) | 4 (8) |
| 60–69 | 32 (29) | 14 (29) |
| 70–79 | 41 (37) | 19 (39) |
| 80–90 | 21 (19) | 10 (20) |
| >90 | - | 1 (2) |
| **Ethnicity** | | |
| Mixed—White & Black African | 1 (1) | - |
| White—British | 111 (99) | 49 (100) |
| **Current Smoker status** | | |
| Ex-smoker | 19 (17) | 5 (10) |
| Current | 28 (25) | 13 (27) |
| Never | 65 (58) | 29 (59) |
| Unknown | - | 2 (4) |
| **Time of urine sample in relationship to death (weeks/month before death)** | Samples analyzed | |
| Week 1 | 26 (23) | 13 (27) |
| Week 2 | 18 (16) | 8 (16) |
| Week 3 | 12 (11) | 5 (10) |
| Week 4 | 2 (2) | 5 (10) |
| Month 2–3 | 13 (12) | 1 (2) |
| Month >3 | 41 (37) | 17 (35) |

[a]*NSCLC* Non-small cell lung cancer; [b]*SCLC* Small cell lung cancer; [c]Based on Multidisciplinary Team discussion.

metabolites with a greater than 2-fold change between different time intervals; the last 4 weeks of life (0–4 weeks), last 2 weeks (0—2 weeks), last 5 days (0–5 days) and last 3 days (0–3 days) (see Supplementary Data File 2 and 3). Some metabolite changes were especially large in the last 3 days, for example, creatine 15-fold and sarcosine 17-fold. In total, we identified 125 metabolites that changed using ANOVA and or volcano plot analyses; 53 metabolites were identified by both approaches. A table of the molecular mass and retention times (RT) for the Unknown metabolites is in Supplementary Table 1.

## Prediction model—training cohort

To focus the model on metabolites that best predicted death we used the Cox Lasso regression approach. This approach offers a powerful framework for survival analysis by simultaneously addressing issues such as variable selection, overfitting, multicollinearity, and interpretability, leading to more accurate and reliable models for predicting survival outcomes. In addition, the Cox Lasso regression approach identifies a minimum set of features that are most relevant for predicting survival outcomes, which can aid in understanding the underlying biological factors associated with outcome (death). Using this approach on our Training cohort ($n = 112$), we derived a multivariable model predicting the probability of survival for each day in the last 30 days. The model utilized only seven metabolites (Table 2), five which increase and two that decrease towards death. The model can be used to assign an individual a risk score indicating their probability of death within the last 30 days. Stratifying subjects into risk groups based on model predictions and plotting their survival curves can reveal a model's discriminatory power. We therefore divided our patients up into three equally sized groups based on their risk of death in 30 days. Kaplan–Meier survival curves were plotted for patients classified as low, medium and high risk of dying (see Fig. 1, log-rank test $p < 0.001$). This approach allows for the visualization and comparison of survival probabilities among individuals grouped into different risk categories based on their predicted likelihood of death within the last 30 days of life, as estimated by the Cox Lasso regression model. The Kaplan–Meier survival curves show it is possible to assign individuals into different risk categories i.e. the differing levels of predicted risk were reflected in the numbers of observed deaths in each group. The group of patients with the highest predicted risk of death, did indeed have the highest numbers of deaths. Conversely, the group with lowest predicted probabilities experienced the fewest deaths. The Low risk group predicted those who are unlikely to be in the last 4 weeks of life; 3% (1/37) died by day 10 and 5% (2/37) by days 20 and 30. The High-risk group predicted those in the last days of life; 59% (22/37) died by 10 days, 89% (33/37) by 20 days and 92% (34/37) by 30 days. The model had excellent Area Under the Curve (AUC) values, which quantifies the overall accuracy of the test, for every day in the last 30 days; for example, 0·85, 0·85, 0·88 and 0·86 on days 5, 10, 20 and 30 (see Fig. 2). We calibrated the model at days 30, 20 and 10 which was reasonable (Supplementary Fig. 11). Calibration is the agreement between the predicted probabilities produced by the model and the actual observed outcomes which is crucial for ensuring the reliability and trustworthiness of predictive models.

Compound identification of unknown metabolites in our model was attempted using public spectral libraries including SIRIUS (which included the HMDB database), Metlin, Mona and mzCloud. This approach did not yield confident chemical structure identifications based on acquired MS/MS fragmentation spectra but did indicate clear molecular formula predictions for Unknown Metabolite 5 ($C_6H_6O_2$) and Unknown Metabolite 7 ($C_{11}H_{18}N_2O_6$). As these are the major spectral libraries in metabolomics research, it suggests these metabolites have not been well-characterized before. Importantly, these unknown metabolites are not medications or medication metabolites. The spectra data for the two unknowns are included in Supplementary Tables 2 and 3. While Cox Lasso Regression analysis identified seven metabolites, five of these metabolites were identified by ANOVA analysis as being significant. ANOVA plots of five of the metabolites identified from the prediction model analysis and compared over the last 12 weeks of life versus >3 months from death are shown in Fig. 3.

**Table 2 | Table of metabolites for the 30-day Cox lasso logistic regression model and the corresponding Hazard Ratio**

| | Metabolite | Hazard Ratios |
|---|---|---|
| 1 | Creatine | 1·66 |
| 2 | Indole-3-lactic acid | 1·59 |
| 3 | Gluconic acid | 0·75 |
| 4 | Carnitine | 1·18 |
| 5 | Histidinyl-hydroxyproline | 0·97 |
| 6 | Unknown Metabolite 5[a] | 1·05 |
| 7 | Unknown Metabolite 7[b] | 1·05 |

[a]Unknown Metabolite 5 ($C_6H_6O_2$) with molecular mass 110.0368 and retention time 4.65.
[b]Unknown Metabolite 7 ($C_9H_{16}N_5O_5$) with molecular mass 274·1148 and retention time 3·48.

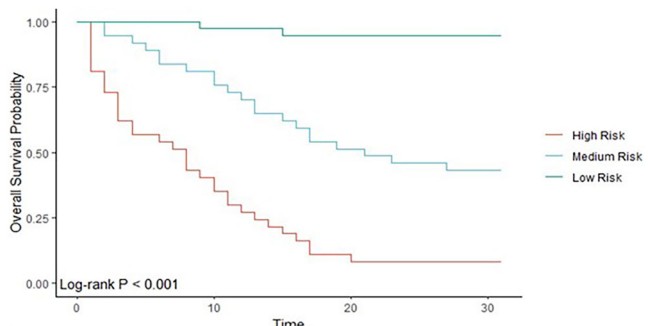

| Percent Survival for each risk group at different time points (number at risk) | | | |
|---|---|---|---|
| | **10 day** | **20 day** | **30 day** |
| **Low Risk (37)** | 97% (36) | 95% (35) | 95% (35) |
| **Medium Risk (37)** | 81% (30) | 51% (19) | 43% (16) |
| **High Risk (37)** | 41% (15) | 11% (4) | 8% (3) |

**Fig. 1 | Kaplan–Meier survival curves using the model separates individuals into High, Medium and Low risk of dying in the Training cohort.** Kaplan–Meier survival curves using the 30 day Cox lasso logistic regression model for the Training cohort showing High, Medium and Low risk of dying. The tables show the percentage survival for the low, medium and high risk of dying groups at 10, 20 and 30 days. The actual numbers alive assigned for each risk category is in brackets. There were initially 37 patients in each group.

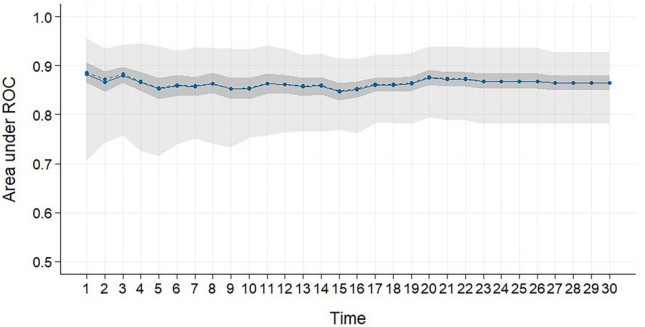

**Fig. 2 | AUC for the 30 day model.** AUC for the 30 day Cox lasso regression model that shows the AUC value at each day for the last 30 days before death. The blue line shows the mean and dotted line the median. The dark grey shows the confidence interval and the light grey shows the minimum and maximum values.

## Prediction model—validation cohort

Predictive modelling relies on identifying patterns within the data that directly relate to the outcome of interest i.e. death. The use of techniques like cross-validation (our validation cohort) ensures the model is robust and can

**Fig. 3 | Plots for each metabolite from the prediction model identified as significant by ANOVA over the last 12 weeks of life versus >3 months from death.** Plots for each metabolite from the prediction model identified as significant by ANOVA (i.e. FDR adjusted $p$-value <0·05.). Each metabolite is compared over the last 12 weeks of life versus >3 months from death. Week 01 ($n = 26$), 02 ($n = 18$) and 03 ($n = 12$) on the $x$-axis indicates patients' measurement in the last week, 2 weeks and 3 weeks of life; week 04+ ($n = 15$) indicates measurements from week 4 to week 11; Week 12+ ($n = 41$) indicates >3 months before death. Data were normalized by reference feature, log-transformed and auto scaled (centered around the mean and divided by the standard deviation of each variable).

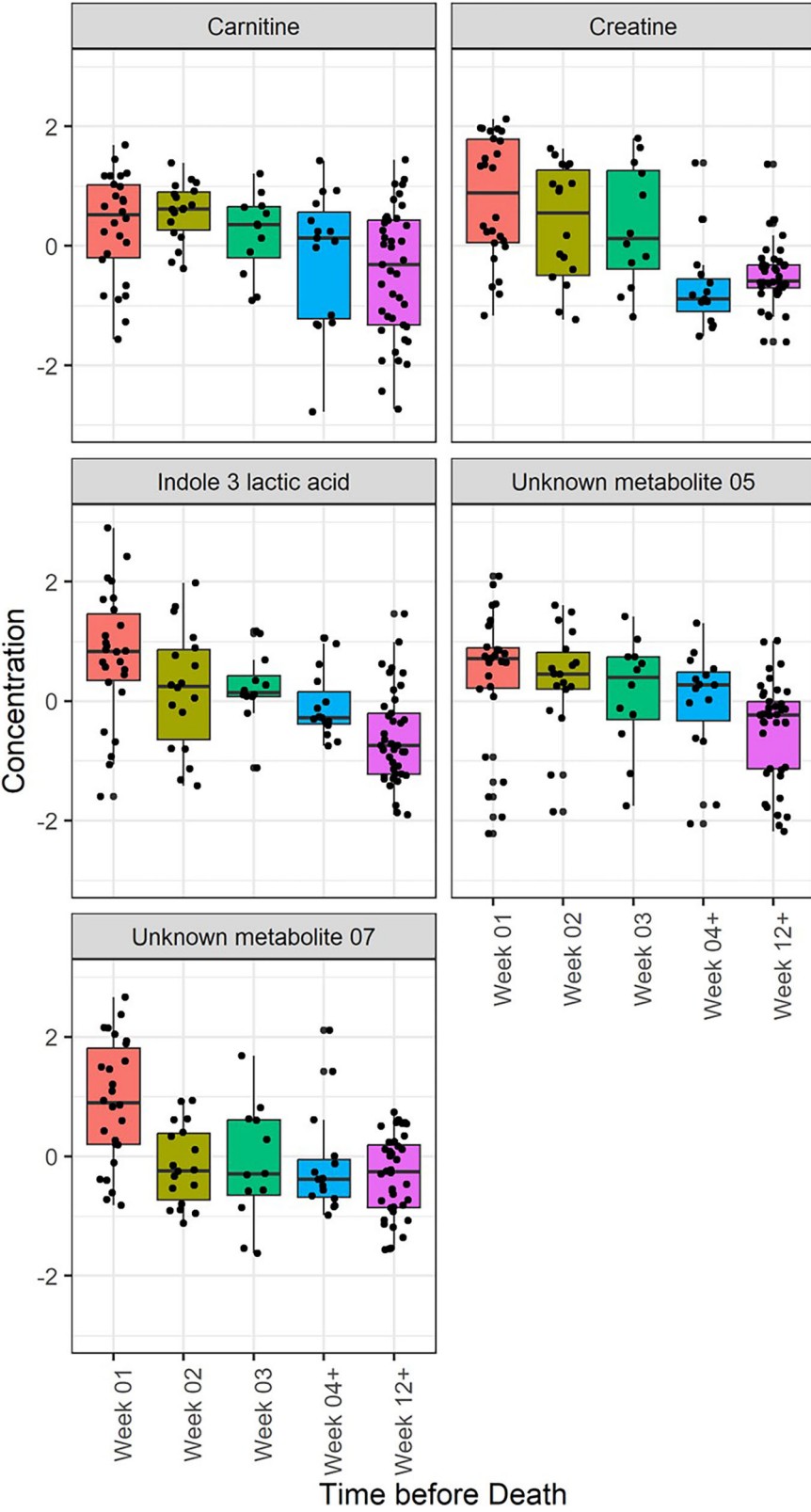

generalize well to new data, eliminating the need for a traditional control group. We validated our model in an independent Validation cohort ($n = 49$) see Table 1. Validating a model generated from a large dataset is crucial to ensure accuracy, generalization, and reliability. It verifies the model's ability to accurately predict unseen data, detects overfitting, aids in model selection, instills trust, and assesses robustness to variations or changes in the system. The Validation cohort was recruited after the initial Training cohort and analyzed 3 years after the Training cohort. Validation of the samples was firstly done on the original LC-QTOF-MS and secondly on a different LC Orbitrap in a different laboratory (cross-laboratory validation). In metabolomics, validating on a different machine is important to ensure the robustness and reproducibility of results. It helps assess whether

the findings are consistent across different instrumentations, reducing the risk of bias or errors specific to a single machine. This validation enhances the reliability and generalizability of metabolomic analyses. The cross-laboratory validation measured creatine (MS/MS)[14], carnitine, gluconic acid, indole-3-lactic acid, unknown metabolite 7 ($C_{14}H_{15}N_3O_3$) and histidinyl-hydroxyproline ($C_{11}H_{14}N_4O_3$) matched by formula, RT and MS/MS spectra similarity. However, unknown metabolite 5 ($C_6H_6O_2$) was not detected. The following results were based on the first validation. Thirty-one patients were in the last month of life. Kaplan–Meier survival curves were plotted for patients classified as low, medium and high risk of dying (see Fig. 4, log rank test $p < 0.001$). The Low risk group predicted those unlikely to be in the last 4 weeks of life; 0% (0/16) died by day 10, 12% (2/16) by days 20 and 25% (4/16) by days 30. The High-risk group predicted those in the last days of life; 56% (9/16) died by 10 days, 87% (14/16) by 20 days and 94% (15/16) by 30 days. A plot of the Area Under the Curve (AUC) for the model for each day over the last 30 days was plotted (see Fig. 5). Time dependent AUC analysis can provide valuable insights into a model, including continuous monitoring of model performance, detection of changes or

deterioration in predictive power, validation of model stability, and identification of shifts in data characteristics or phenomena being modelled over different time periods. The 30 day model had excellent AUC values for every day in the last 30 days; for example, 0·86, 0·83, 0·90 and 0·86 on days 5, 10, 20 and 30. Calibration of the model at days 30, 20 and 10 was good (Supplementary Fig. 11). Despite what can be considered a small number of samples in the last month of life for both the Training and Validation cohorts our results demonstrate the robustness and reproducibility of our model.

## Pathway analysis

Finally, we explored pathways affected during the dying process. The metabolites identified from the Training dataset are compatible with a reduction in the rate at which molecules are synthesized or produced within biosynthetic pathways and, in the case of degradative products, increased breakdown of pathway intermediates. The total number of metabolites identified, the number of these identified by KEGG and the number of metabolites used in the pathway analysis for the last 2 weeks and last 3 days from the different LC-QTOF-MS analysis approaches is shown in Supplementary Table 4. KEGG is a widely used database and resource for understanding the molecular functions and biological systems of organisms, including biochemical pathways. KEGG pathway analysis, using the statistically significant metabolites discovered for the last 2 weeks and last 3 days of life, identified several associated biochemical pathways (Supplementary Table 5). Pathways involved in the last weeks of life are summarized in Table 3 and in further detail in Supplementary Data 4. These pathways are collated from the metabolites identified as changed significantly from the volcano plot analysis and ANOVA analysis of the Training dataset.

## Discussion

Our work describes an objective model predicting dying based on urinary metabolites and represents to the best of our knowledge, the first attempt using a metabolomic approach to describe pathways affected during the dying process. The model was validated on an independent cohort of patients and found to be reproducible. Using a risk score, we were able to categorize patients at risk of dying. A test predicting the last days of life is important; by decreasing clinical uncertainty, it will support clinical practice and improve the care of dying patients[28], enabling families, medical teams and health-care providers to plan and provide the best care possible.

The Cox Lasso derived model demonstrates it is possible to use urine metabolites to predict the dying process for each day within the last 30 days of life with good accuracy in both the Training and Validation cohorts. Using the 7 metabolites identified, the model can assign an individual a risk score indicating their probability of death within the last 30 days. The High-risk of dying score predicted the majority of patients imminently dying. The

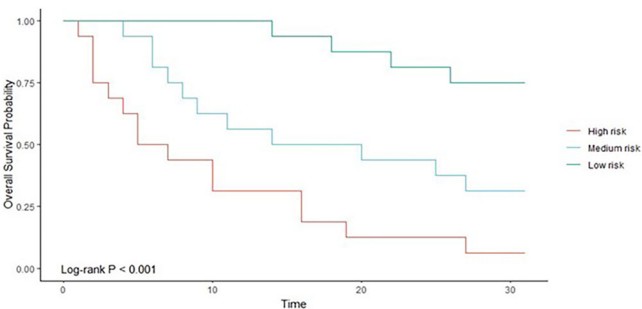

| Percent Survival for each risk group at different time points (number at risk) | | | |
|---|---|---|---|
| | **10 day** | **20 day** | **30 day** |
| **Low Risk (16)** | 100% (16) | 88% (14) | 75% (12) |
| **Medium Risk (16)** | 63% (10) | 50% (8) | 31% (5) |
| **High Risk (16)** | 44% (7) | 13% (2) | 6% (1) |

**Fig. 4 | Kaplan–Meier survival curves using the model separates individuals into high, medium and low risk of dying in the Validation cohort.** Kaplan–Meier survival curves using the 30 day Cox lasso logistic regression model for the Validation cohort showing High, Medium and Low risk of dying. The tables show the percentage survival for the low, medium and high risk of dying groups at 10, 20 and 30 days. The actual numbers alive assigned for each risk category is in brackets. There were initially 16 patients in each group.

**Fig. 5 | Comparison of AUC for training and validation cohorts.** The time dependent Area Under the ROC Curve comparing the Training and both Validation cohorts (same laboratory and different laboratory) for each day in the last 30 days of life. One validation cohort was validated on the same LC-QTOF-MS in the same laboratory as the Training cohort. The second validation cohort was validated on a LC Orbitrap MS in a different laboratory.

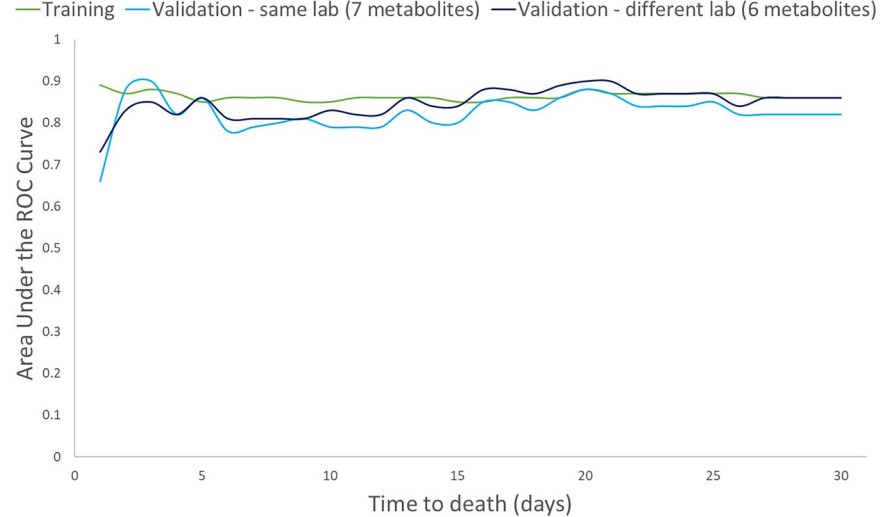

## Table 3 | Pathways involved in the last weeks of life

| Pathways involved in the last weeks of life |
| --- |
| Energy metabolism |
| Disrupted mitochondrial fatty acid β-oxidation |
| Mitochondrial dysfunction |
| Decreased RNA synthesis |
| Decreased protein synthesis |
| Altered 1-carbon metabolism |
| Altered nucleoside (purine, pyrimidine) synthesis |
| Muscle loss/damage |
| Oxidative stress |
| Cell membrane breakdown / turnover |
| Altered hormone production |
| Altered amino acid metabolism |
| Kynurenine pathway activation |
| Decreased oral intake |

This table is a summary of all the metabolites identified as changed significantly from the volcano plot analysis and ANOVA analysis of the Training dataset. Further information to support this table is in Supplementary Data File 4.

Low-risk of dying score identified those not in the last 2 weeks of life and at 30 days 75% of the cohort survived. The model calibration was stable at 30, 20 and 10 days; slightly overestimating those likely to die with a High-risk score and underestimating dying with a Low-risk score. External validation in an independent cohort had excellent AUC values. To our knowledge, it is the only model predicting dying across a range of time points including the last 2 weeks, with an AUC of 0·86 and 0·83 on days 5 and 10 in the Validation cohort. Further work with increased numbers of patients and samples will calibrate our model closer to dying and potentially >30 days.

A recent comparison of five validated end-of-life prognostic tools[29] showed the best model, PiPS-B (based on clinical observations and blood results) for 14 day and 56 day prognostication, was as accurate as expert multidisciplinary clinician judgement; the overall accuracy was 61%. The PiPS-A model (clinical observations only) obtained a C-statistic of 0.825 (0.803–0.848) and the PiPS-B model obtained 0.837 (0.810 – 0.863). Whilst not the same as the time-dependent AUC we calculated (which accounts for censoring of observations) it is very similar. Our model obtained slightly higher AUCs in both the Training and Validation cohorts and uniquely gave accurate prediction across a range of time points (between 1 and 30 days).

We do not know how people die from cancer. No prior study has used a metabolomics or any other –omics approach to the best of our knowledge to investigate dying from cancer. Our work therefore is the first study describing metabolic pathways involved or affected in the last weeks and days of life and thus provide insights into the dying process. The pathways identified are compatible with a reduction in the rate at which molecules are synthesized or produced within biosynthetic pathways and, in the case of degradative products, increased breakdown of pathway intermediates. The Cox Lasso regression approach taken aids in understanding the underlying biological factors associated with death and our model identified 7 metabolites suggesting the involvement of different pathways. Creatine, which increases nearly 15 fold in the last 3 days of life and carnitine are both muscle metabolites suggesting increasing muscle damage or cancer related cachexia. Carnitine may also indicate disrupted mitochondrial fatty acid β-oxidation and or mitochondrial dysfunction. Gluconic acid, abundant in plants, decreases and suggests decreased oral intake. Indole-3-lactic acid is a metabolite of tryptophan, metabolized by two major pathways in humans, either through kynurenine or via a series of indoles; indole derivatives are known to have anti-inflammatory roles. Histidinyl-hydroxyproline, a likely breakdown product of the dipeptide histidylhydroxyproline, is a metabolite identified from the in vitro growth of osteoclasts on dentine. It results from

the breakdown of bone collagen, suggesting bone resorption pathways are affected.

ANOVA and volcano plot analysis identified a range of additional metabolites that change in the last weeks and days of life. Towards the end-of-life there are changes in energy availability. Within muscle, increased creatine and decreased creatinine levels suggest a shortage of available phosphocreatine, likely resulting in a shortage of available high energy phosphate (ATP). Increased NAD+ and disrupted mitochondrial fatty acid β-oxidation further supports the notion of reduced energy availability. Disrupted mitochondrial fatty acid β-oxidation is indicated by increased urinary dicarboxylic acids. Mitochondrial dysfunction is suggested by increased urinary carnitine (essential for fatty acid metabolism), methyl-glutaric acid and hydroxyl-3-methyl-glutaric acid. Interestingly, multiple inflammatory mediators are known to affect mitochondrial energy metabolism and mitochondrial dynamics, in turn mitochondrial dysfunction can promote inflammation[30]. We previously demonstrated an increase in acetone, produced during mitochondrial fatty acid β-oxidation, towards the end of life[31]. In addition, changes in xanthine metabolism (increased xanthine and hypoxanthine) are well known to be associated with oxidative stress.

Numerous changes suggest altered nucleic acid metabolism. Several critical building blocks accumulate, in particular UMP essential for RNA synthesis, as well as adenine and guanosine. In addition, purine degradation products xanthine, and hypoxanthine increase. Collectively these suggest depleting pools of substrates required for nucleic acid anabolism. Crucially, this would suppress ribosomal biogenesis, impacting upon cellular capacity for protein synthesis, and importantly cellular stress monitoring and cell viability[32]. In addition, ribosome synthesis makes high demands on cellular energy resources which appear to be low given the increase in NAD+. Other changes also likely impact nucleic acid metabolism; alterations in one-carbon metabolism, indicated by accumulating dihydrofolic acid, sarcosine, and cystathionine, suggest a reduced ability to generate nucleotides for DNA synthesis. In addition, there will be reduced capacity to deal with reactive oxygen species through decreased production of sulfhydryl-containing reducing agents. Altered one-carbon metabolism is highly associated with aging[33], however, the consequences of changes in these processes are difficult to predict[34].

There was evidence of muscle damage or breakdown and an increase in amino acids starting ~3 weeks before death. Interestingly, increased muscle protein breakdown and efflux of amino acids is a fundamental response seen in critical illness[35]. Numerous hormones, including their essential intermediates, were altered; cortisol, epinephrine, histamine and hydroxytryptophan increase; dopamine and serotonin decrease. Approximately 5% of tryptophan is converted to serotonin, therefore the low serotonin levels and increased kynurenine levels imply there is a diversion[36]. Cortisol was previously shown to be increased in dying patients including a cohort of patients with lung cancer[37]. In critically ill patients, increased cortisol was shown to be related to decreased breakdown in the liver[38]. Both cortisol and tryptophan metabolism are influenced by the circadian rhythm yet in the last days of life cortisol production and tryptophan metabolism increase. Taken together, the pathways involved or affected during dying imply the influence of cancer on the body. Considering the similar trajectory of dying observed in individuals with cancer[39] our findings are likely applicable to other types of cancer.

An advantage of our study is that it was based on a homogenous cancer population of lung cancers. Our Validation cohort was recruited and analyzed after the Training cohort. Our objective model is only based on seven urinary metabolites and it prognosticates for every day within the 30 days of life with excellent AUC values. The model validated very well. To our knowledge, it is the only model that predicts dying across a range of time points (between 1 and 30 days). We believe it is an advantage that diet was not controlled for. Our analysis identified 10 food related metabolites that decreased with statistical significance in the last weeks of life. Therefore decreased oral intake was identified as a pathway involved in dying in Table 3. Importantly, our analysis showed a decrease in these metabolites

concentration in the last weeks compared to those >3 months from death, that decreased further in the last week of life.

The structure of two metabolites in our model remain unknown despite an extensive search against an in-house library (619 metabolites) and public spectral libraries such as: SIRIUS, Metlin, Mona and mzCloud. These are the major spectral libraries in metabolomics suggesting these metabolites have not been well characterized before. Importantly, medications and medication related metabolites were excluded. For those samples, predominantly collected in months before death, patients were admitted to hospital. The main reason a patient with advanced cancer was admitted to hospital was for an infection and therefore these patients would have been on IV antibiotics, predominantly Tazocin. For those patients in the last days of life, the majority were not on antibiotics. Therefore, an increase in antibiotics metabolites in the last days of life would not be detected. In the last days of life, patients normal medications would have stopped due to decreased oral intake or stopped for clinical reasons. Therefore, the metabolites of non-symptom related medications (e.g. antihypertensives etc) would be expected to decrease. However, certain symptoms can increase towards the end of life. To manage these, medications would either be added or possibly increase. In general there is a limited number of medications used to control symptoms at the end of life e.g. morphine for pain, midazolam for terminal agitation, glycopyrronium for respiratory secretions. The well characterised metabolites of these medications were not detected in our analyses. The fact the unknown metabolites described were present in almost all samples makes a drug artefact possibility unlikely.

There are some limitations to our study. Our Training and Validation cohorts were based on relatively small sample sizes[40] (Training $n = 112$, Validation $n = 49$) and the limited number of samples in the last month of life (Training $n = 58$, Validation $n = 31$) excluded longitudinal analysis. Ideally, a time series on a group of individuals each sampled repeatedly over several weeks would give the best models. However, it is very difficult predicting time of death in the last weeks of life, and we do not know who is going to die and when. We did analyse the samples of patients over time from the few patients we had collected however statistical analyses were consistently underpowered and we have not presented this data. Therefore, our pragmatic approach was to sample all patients with advanced disease regardless of perceived prognosis and use a cross-sectional approach to identify patterns or predictors of dying. Our population was homogenous and therefore not a broad cultural cross-section of patients. Furthermore, two of the metabolites included in the model remained unknown despite extensive searches against in-house and public spectral libraries. In addition, it is important to note our analysis was based on urine measurements. While convenient, as the sample collection was non-invasive, it is well understood that urine offers a limited view for biochemical pathway analysis in comparison to blood. Moreover, the hazard ratios for the model predicting dying were derived from semi-quantitative data[41] i.e., the Training and Validation datasets relied on relative compounds intensities, not exact metabolite concentrations. While such measurements make possible the exploration of relative differences within a study, exact quantitative measurements are needed to allow for evaluation and prediction on new patients. An important part of translating this model will be to establish concentration reference ranges for these metabolites in dying versus not-dying patients. Finally, we must note that empirically there are different time frames of dying, which is an added complication to predicting the last days of life. When patients with cancer are recognized to be dying (actively dying) in a hospice setting, they usually die over two to three days. However, some die within 24 h and some take longer than a week. By recruiting greater numbers of patients with lung and other solid tumours in the last weeks of life and identifying those cohorts that die in these different ways, we may improve the accuracy of our model. Further work is needed to develop a widely applicable robust clinical tool.

Our work describes an objective validated model predicting dying based on urinary metabolites. It also represents the first attempt using a metabolomic approach to describe metabolites that change before death, thereby providing insights into biochemical pathways involved in or affected by the dying process. They imply not only the influence of cancer on the body but also illustrate the dying process. Considering the similar trajectory of dying observed in individuals with cancer, our findings are likely applicable to other types of cancer. We are exploring the development of a low cost commercial point of care diagnostic test system. Accurate prognostic information at the end of life is essential to co-ordinate and manage care in response to need, whilst avoiding burdensome and unnecessary interventions. The early recognition a person may be dying is central to all the priorities for improving peoples' experience of care in the last days and hours of life. Prognostic tests, based on the metabolites identified in this study, could aid in the early recognition of people who may be dying, and therefore have the potential to influence clinical practice and improve the care of dying patients substantially.

## Data availability
The raw data and .csv files from the LC-QTOF-MS analyses in the paper are uploaded on the Metabolomics Workbench database under Study ID ST002082 and available at https://doi.org/10.21228/M8TT44[42]. Source data is located in Supplementary Data 5.

## Abbreviations

| | |
|---|---|
| ANOVA | Analysis of variance |
| AUC | Area under the receiver operating characteristic curve |
| CD | Compound discoverer |
| FC | Fold change |
| FDR | False discovery rate |
| FWHM | Full Width Half Maximum |
| KEGG | Kyoto Encyclopaedia of Genes and Genomes |
| LASSO | Least absolute shrinkage and selection operator |
| MS | Mass spectrometry |
| MS/MS | Tandem mass spectrometry |
| MSI 1 | Non-tandem mass spectrometry (collision energy not applied) |
| m/z | Mass-to-charge ratio |
| PQN | Probabilistic quotient normalization |
| UHPLC | Ultra-high performance liquid chromatography |

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

## Acknowledgements

We would like to acknowledge funding from the Wellcome Trust Seed award for Science, 202022/Z/16/Z (S.C.), North West Cancer Research award, SI2018.11 (S.C., J.E.E., C.P., M.B.), University of Liverpool Enterprise Investment Fund award (SC, CP) and Novo Nordisk Foundation, NNF20CC0035580 (DBK). We would like to thank: the Liverpool AKU Research Group (Liverpool University Hospitals Foundation Trust) & Prof. Ranganath for spectrometer access; The research nurses who recruited patients and collected samples (F Westwell, C Harrop, S Stanley, P Walker, S Barnes, G Hull, S Dealing, N Collins, M Leach); the staff of the Liverpool University Hospitals Foundation Trust (including Heather Rogers), The Clatterbridge Cancer Centre, Marie Curie Hospice Liverpool, Willowbrook Hospice, Prescot, St Helens and Knowsley Hospitals Trust (including Jeanette Anders and Paula Scott), St Catherines Hospice, Preston, Trinity Hospice, Blackpool; the principal investigators for the study (Dr J Bellieu, Dr A Thompson, Dr A Pope, Dr L Chapman, Dr A Fletcher, Dr A Gadoud); and to E Wright and B Hughes our lay members for their invaluable input.

## Author contributions

All authors were involved in critical review of the manuscript and have seen and approved the final version. Specific contributions as follows; Study conception and design: S.C., E.C., S.M., C.R.M., A.C.N., M.T.B., J.E.E., C.P. Sample acquisition: S.C., C.R.M., S.M., J.E.E. Sample analyses E.C., R.S., B.P.N., A.S.D., I.R., D.B.K. Data analysis: E.C., J.B., R.S., B.P.N., A.S.D., I.R., D.B.K., S.C., D.M.H. Supervision: S.C., C.P., L.R.R., J.A.G., D.B.K. Drafting

the manuscript: S.C., E.C. and C.P. Revision of manuscript: S.C., E.C., R.S., B.P.N., A.S.D., I.R., J.B., D.M.H., A.C.N., C.R.M., S.M., M.T.B., D.B.K., J.E.E., C.P.

## Competing interests

The authors declare the following competing interests: a patent application was submitted March 2022, UK Patent Application No GB2204213.9, Biology of dying (SC, CP). All other authors declare no competing interests.
