## [Transparent peer review file · Communications Medicine]

Reviewers' comments:

Reviewer #1 (Remarks to the Author):

This manuscript describes an investigation into urinary metabolites to predict the dying process in lung cancer patients. The authors carefully searched for metabolic changes and attempted to identify meaningful markers that could potentially be used by clinicians for better medication and decision-making. The research is interesting and may provide insights not only for predicting the dying process but also for identifying early diagnostic markers for lung cancer patients. After revision, this manuscript can be considered for publication.

Comments:

1. In Table 1, the authors divided the cohort according to sex, diagnosis, age, and so on. However, urinary metabolic alterations can be affected by medication, cancer treatment methods, cancer stage, and other factors. Therefore, it would be better to address how this kind of patient heterogeneity can be controlled.
2. Figure S1 is too long and hard to identify systematically. It is recommended to combine the data. Additionally, the letter size is too small to read.
3. The investigated dying period is divided into groups for each week and for more than 4 weeks. The authors should address why a weekly difference in dying prediction is important and meaningful. Various variables can influence the dying process at the end of life.
4. Figure 1C seems to be missing.
5. It is recommended to enlarge the letter size, especially for graphs.
6. If possible, it is recommended to discuss the utilization of diagnostic metabolic biomarkers (found in this work) for early lung cancer detection.

Reviewer #2 (Remarks to the Author):

This article of Coyle et al describes urinary metabolomics analysis on patients with lung

cancer to create a metabolite model to predict dying over the last 30 days of life. The model showed accuracy in a Training cohort and Validation cohort using 7 metabolites and to give accurate predictions over a range of time points in the last 30 days of life. Additionally, they present findings on pathways involved in or affected by the dying process. Prognostic tests, based on the metabolites identified, could aid clinicians in the early recognition of people who may be dying and thereby influence clinical practice and improve the care of dying patients.

Overall, this is a very well written and well composed paper on original research. Compliments for including a training and validation cohort in a different laboratory, albeit of relatively small size, showing the reproducibility of the study as well as cross-laboratory validation. The topic of predicting the dying process in cancer patients is interesting and relevant for improving the care of dying patients as well as guiding the patient and relatives in the process. This study is especially interesting, as it uses urine for analysis, which lowers the burden in this frail population.

Three major points of attention/clarification from the point of view of a clinician:

1. For a reader not involved in this modelling it's not clear why there is an equal distribution of patients between low - intermediate - high risk in both the Training and Validation cohort. Also in both groups one patient is missing, which is as far as I have seen not clarified in the text.
2. It is a disadvantage that there is only one urine sample per patient. It's not clear why more than one sample per patient will bias the results as is stated in the methods. Also, it seems more appropriate to collect samples over time in one patient at the end of life to see possible changes in metabolites at the end of life.
3. It would be interesting to elaborate on how this would be implemented into future care (when developed into a widely applicable robust clinical tool) as it would be quite expensive and especially time consuming to analyze every patient with LC-QTOF-MS.

Some additional suggestions:

1. Please provide Hazard Ratios with 95%CI when possible
2. Is it known if there is any difference in outcome between different diagnoses, age ranges and smoker status? If so, should the data be adjusted for this difference?
3. Some editing is needed on the introduction. For example: the second sentence of the second alinea better connects to the first alinea than the first sentence of the second alinea.
4. Please clarify why has been chosen for lung cancer patients, and why for only lung cancer patients.

Still, this article offers a first look at what is possible to predict around the end of life and as such valuable and an motivation for more research. I highly recommend publication.

Reviewer #3 (Remarks to the Author):

The manuscript titled "Urinary Metabolite Model to Predict the Dying Process in Lung Cancer Patients" presents a novel and innovative approach to understanding the biochemical changes occurring in the final stages of life in individuals with lung cancer. This study use of urinary metabolomics to develop a predictive model for the dying process, offering a non-invasive, practical method for clinicians and researchers. However the reviewer has some concerns.

Major Comments:

Study design

1. Clarity: The manuscript lacks clarity in the study design section. It is difficult to follow how the study was performed.
2. Study population: Very little information about the study group is presented, with no inclusion or exclusion criteria. Potential confounding factors such as age, sex, therapy (chemo therapy or radiation therapy) and diet are not adequately addressed. These factors could significantly influence the urine metabolome and should be accounted for in the analysis. In addition some of these metabolites might be associated with lung cancer and may not be specific to the dying process. This needs to be clarified or refocused to ensure relevance to the study's main objective.
3. Control Groups: The study appears to lack appropriate control groups. It would be beneficial to compare this new method with current methods and include control samples from healthy individuals. This would help in establishing the specificity and relevance of the findings.
4. Time Points: The study seems to include four time points when refereeing to Table 3 however, however according to line 482 "In order not to bias the results, only one sample per patient, the one closest to death, was included for analysis." Please clarify. In addition, the study seems to have very few data points which may be insufficient for capturing the full dynamics of the dying process. Using more time points could provide more comprehensive insights, and the reviewer doesn't understand the risk of bias?

Interpretation

5. Interpretation of Results: The interpretation of the results, particularly regarding false

positives, i.e. when the prediction was incorrect, is not clear. The manuscript should discuss why certain individuals lived longer or shorter than predicted based on the metabolome data.

6. Circadian Rhythm and Analyte Measurement: The manuscript does not discuss whether the analytes exhibit circadian rhythms that might be influenced by factors such as diet. It is important to specify when the measurements were taken to ensure consistency and accuracy.

Method

7. Calibration and Batch Effects: The manuscript mentions calibration points but does not clearly explain how they were used. Furthermore, it is stated that samples were run in batches, but it is unclear how batch effects were corrected. More details on the calibration process and batch effect correction are needed to assess the reliability of the results.

8. Group Creation: The criteria for creating the three groups mentioned in the study are not described. Clear definitions and justifications for group divisions are necessary for understanding the comparative analyses.

9. Figure 2: There appears to be a discrepancy in Figure 2, with two metabolites missing. This needs to be addressed and corrected. Figure 2 suggests that all metabolites decrease, which could be due to dietary differences and reduced appetite. This possibility should be explored and discussed.

10. Table 3 Metabolites: It is unclear which metabolites Table 3 is based on. The manuscript should specify whether it includes all metabolites and all participants or only a subset for example the test set. Additionally, information on how the analytical quality of the data was ensured is required.

11. Exclusion of Medication Metabolites: The manuscript states that medications and related metabolites were excluded. It is important to consider that some of these unidentified metabolites could be drug artifacts. A more thorough explanation of the exclusion criteria and potential implications is needed.

12. While the study is described as prospective, it is not clear whether all samples were analyzed at the same period after collection. This is further complicated by having two “separate” method sections with differences in amount of information.

Minor Comments:

1. Sample Preparation and Storage: There is insufficient information regarding the preparation and storage of samples before analysis. Detailed descriptions of these processes are essential for reproducibility and understanding potential sources of variability.

2. HMDB Database: Consider using the HMDB database for metabolite identification, as it

is a well-established resource that could enhance the study's reliability and comparability.

3. Abbreviations: The manuscript uses abbreviations such as LC-QTOF and LC-Orbitrap. Both instruments are MS/MS systems, but the abbreviation MS/MS is often associated with triple quad instruments rather than high-resolution instruments like Orbitrap. It is recommended not to abbreviate Orbitrap with UHPLC-MS/MS, as this can be misleading. Overall, the manuscript requires significant revisions to address these major and minor concerns. Improving clarity in the study design, providing detailed methodological information, and including appropriate controls will strengthen the study's validity and impact.

Response to Reviewers' comments

July 2024

Reviewer #1

Comments:

1. In Table 1, the authors divided the cohort according to sex, diagnosis, age, and so on. However, urinary metabolic alterations can be affected by medication, cancer treatment methods, cancer stage, and other factors. Therefore, it would be better to address how this kind of patient heterogeneity can be controlled.

For our study, a traditional control group was not appropriate. This study is a study of people with advanced cancer and observing patterns and predictors of death. Predictive modelling relies on identifying patterns within the data that directly relate to the outcome of interest i.e. death. The use of techniques like cross-validation (our validation cohort) ensures the model is robust and can generalize well to new data, eliminating the need for a traditional control group.

Discussing this further:

In many respects the above factors cannot be controlled for. The inclusion criteria for the study included people with 'advanced lung cancer' meaning they had incurable lung cancer (locally advanced or metastatic) as determined by a multidisciplinary team meeting. We only recruited lung cancer patients in an attempt to decrease confounding factors. However, we believe the heterogeneity of our samples is an important strength of our study.

The machine learning approach (cox lasso regression analysis) we undertook was able to analyse our samples and even though there was heterogeneity in the samples, the analysis was able to develop a validated model predicting dying. The known metabolites identified in the model are also biochemically justifiable (decreased food intake, muscle loss/cachexia, mitochondrial dysfunction, bone loss/resorption).

In addition, our approach was used by Gwilliam et al used to develop the Prognosis in Palliative care (PIPS) predictor model of dying (<https://www.bmj.com/content/343/bmj.d4920>).

We added the following to the 'Prediction model - Validation cohort' section in the Results:

- *Predictive modelling relies on identifying patterns within the data that directly relate to the outcome of interest i.e. death. The use of techniques like cross-validation (our validation cohort) ensures the model is robust and can generalize well to new data, eliminating the need for a traditional control group.*

Re Medications:

Medications would definitely have metabolites in the urine – in particular there will be the metabolites of the medications themselves. While our work was untargeted, we had a targeted library that did not include medication metabolites. For those metabolites with an unknown structure, medication metabolites were excluded. We wrote in second paragraph of the ‘Prediction model – Training cohort’ section in the Results of the paper:

- *Compound identification of unknown metabolites in our model was attempted using public spectral libraries including SIRIUS, Metlin, Mona and mzCloud. This approach did not yield confident chemical structure identifications based on acquired MS/MS fragmentation spectra but did indicate clear molecular formula predictions for Unknown Metabolite 5 (C₆H₆O₂) and Unknown Metabolite 7 (C₁₁H₁₈N₂O₆). As these are the major spectral libraries in metabolomics research, it suggests these metabolites have not been well-characterized before. Importantly, these unknown metabolites are not medications or medication metabolites.*

We included the following in the ‘Setting and Participants’ paragraph of the Methods:

- *The list of medications a patient was on for the previous 24 hours was recorded for each sample.*

To discuss your comment further, we added the following to 9th paragraph in the Discussion:

- *For those samples, predominantly collected in months before death, patients were admitted to hospital. The main reason a patient with advanced cancer was admitted to hospital was for an infection and therefore these patients would have been on IV antibiotics – predominantly Tazocin. For those patients in the last days of life, the majority were not on antibiotics. Therefore, an increase in antibiotics metabolites in the last days of life would not be detected. In the last days of life, patients normal medications would have stopped due to decreased oral intake or stopped for clinical reasons. Therefore, the metabolites of non-symptom related medications (e.g. antihypertensives etc) would be expected to decrease. However, certain symptoms can increase towards the end of life. To manage these, medications would either be added or possibly increase. In general there is a limited number of medications used to control symptoms at the end of life e.g. morphine for pain, midazolam for terminal agitation, glycopyrronium for respiratory secretions. The well characterised metabolites of these medications were not detected in our analyses. In addition, the fact the unknown metabolites described were present in almost all samples makes a drug artifact possibility unlikely.*

Re chemotherapy and cancer staging, we added the following to the 'Patients' paragraph in the Results:

- *To decrease confounding factors this study only recruited lung cancer patients. TNM staging data were not collected in this study as these are used at cancer diagnosis to predict 5-year survival and guide clinician treatment. Only 3 patient samples analysed were on some form of chemotherapy (afatinib, pembrolizumab and exemestane) and none of these samples were in the last 6 weeks of life.*

2. Figure S1 is too long and hard to identify systematically. It is recommended to combine the data. Additionally, the letter size is too small to read.

We reordered the graphs.

3. The investigated dying period is divided into groups for each week and for more than 4 weeks. The authors should address why a weekly difference in dying prediction is important and meaningful. Various variables can influence the dying process at the end of life.

We are uncertain what you mean by 'various variables can influence the dying process at the end of life'. We highlight in the paper that the recognition of dying is difficult and we made an argument that an objective model estimating risk of death over a range of time periods including the last days of life is needed.

When we started out this work we believed that the most important time period to predict was the last days of life. When we started to look at metabolite changes over time we noticed changes over the last 3 weeks from death. Our risk prediction model looks at risk over a 30 day period, and can predict risk of death at any time point.

However, we arbitrarily divided the time points at the end of life into weeks as a useful way of distinguishing various time periods.

An oncologist is particularly interested in knowing if someone is within the last 4 weeks of life. Subsequent work shows our model not only works on a small cohort of mixed solid cancers it also predicts the last 6 weeks of life (unpublished). This is an important timeframe for an oncological treatment decision making perspective.

From a palliative care perspective, knowing the last weeks of life is important.

- The early recognition that a person may be dying underpins all the priorities for improving people's experience of care in the last days and hours of life.
- A major challenge to providing care is the reliable recognition of dying. Recognising the last weeks of life:
 - makes it possible for patients to come to terms with dying, support where they choose to die and facilitate addressing financial and personal affairs.
 - ensures the timely implementation of necessary conversations, care and treatment
 - prevents futile investigations and inappropriate treatments.

- Allows healthcare systems to create services to plan and provide appropriate care.

We already wrote in the discussion:

- *A test predicting the last days of life is important; by decreasing clinical uncertainty, it will support clinical practice and improve the care of dying patients¹², enabling families, medical teams and health-care providers to plan and provide the best care possible.*
- *Accurate prognostic information at the end of life is essential to co-ordinate and manage care in response to need, whilst avoiding burdensome and unnecessary interventions. The early recognition a person may be dying is central to all the priorities for improving peoples' experience of care in the last days and hours of life. Prognostic tests, based on the metabolites identified in this study, could aid in the early recognition of people who may be dying, and therefore have the potential to influence clinical practice and improve the care of dying patients substantially.*

For clarity, we added the following line into the second paragraph of the Introduction:

- *Accurate prognostic information at the end of life is essential to co-ordinate and manage care in response to need, whilst avoiding burdensome and unnecessary interventions.*

4. Figure 1C seems to be missing.

Apologies. Now included.

5. It is recommended to enlarge the letter size, especially for graphs.

We edited the graphs.

6. If possible, it is recommended to discuss the utilization of diagnostic metabolic biomarkers (found in this work) for early lung cancer detection.

The ability to detect early lung cancer is technically feasible using the metabolomics approach we used. Our study is not suitable to finding biomarkers of early cancer as there was no cancer-free control group, and since our samples were from patients with incurable cancer, we had no samples representative of early stage cancer.

The detection of early cancer is important and a national strategy, with significant funding in the UK is underway to investigate this. Despite this approach, people will always die from cancer and yet laboratory research investigating end of life is non-existent. We believe that discussion of the utilization of diagnostic metabolic biomarkers for early lung cancer detection, while important, to be a distraction from what we consider the important narrative of the paper.

Reviewer #2

Three major points of attention/clarification from the point of view of a clinician:

1. For a reader not involved in this modelling it's not clear why there is an equal distribution of

patients between low - intermediate - high risk in both the Training and Validation cohort. Also in both groups one patient is missing, which is as far as I have seen not clarified in the text.

We added the following line into the 'Prediction model – Training cohort' section in the Results section of the paper:

- *Stratifying subjects into risk groups based on model predictions and plotting their survival curves can reveal a model's discriminatory power. We therefore divided our patients up into three equally sized groups based on their risk of death in 30 days. Kaplan-Meier survival curves were plotted for patients classified as Low, Medium and High risk of dying (see Figure 1 A, log-rank test $p < 0.001$). This approach allows for the visualization and comparison of survival probabilities among individuals grouped into different risk categories based on their predicted likelihood of death within the last 30 days of life, as estimated by the Cox Lasso regression model. The Kaplan-Meier survival curves show it is possible to assign individuals into different risk categories *i.e.* the differing levels of predicted risk were reflected in the numbers of observed deaths in each group. The group of patients with the highest predicted risk of death, did indeed have the highest numbers of deaths. Conversely, the group with lowest predicted probabilities experienced the fewest deaths.*

2. It is a disadvantage that there is only one urine sample per patient. It's not clear why more than one sample per patient will bias the results as is stated in the methods.

In metabolomics studies, analyzing only one sample per subject, especially in large datasets with variable numbers of samples from different subjects, can help in reducing bias (unequal sampling and non-independence of observations) and ensure the reliability of the results.

There are several reasons for this:

1. **Avoiding Pseudoreplication:** When multiple samples from the same subject are treated as independent data points, it can lead to pseudoreplication. Pseudoreplication artificially inflates the sample size and can result in misleading statistical conclusions. Analyzing only one sample per subject helps in treating each subject as a single data point, thus avoiding this issue.
2. **Reducing Intra-Subject Variability:** Multiple samples from the same subject can exhibit variability due to biological or technical factors. By choosing a single representative sample per subject, the analysis can focus on inter-subject variability rather than intra-subject variability, which is often the main interest in biological studies.
3. **Simplifying Statistical Analysis:** Statistical methods used in metabolomics, such as multivariate analysis, are more straightforward and interpretable when each subject contributes a single data point. This simplifies the modeling and interpretation of the results.
4. **Balancing Sample Size:** In studies with an unequal number of samples per subject, analyzing only one sample per subject ensures a balanced dataset. This balance is crucial for statistical methods that assume equal representation of subjects, thereby reducing potential bias introduced by over-representation of subjects with more samples.

5. **Ensuring Independence:** Analyzing multiple samples from the same subject can violate the assumption of independence, which is a key requirement for many statistical tests. Using one sample per subject helps maintain the independence of observations, leading to more robust and valid statistical inferences.
6. **Focus on Biological Variation:** The primary goal of many metabolomic studies is to understand biological variation between subjects, such as differences due to disease states, treatments, or other factors. By analyzing one sample per subject, researchers can more accurately capture this inter-subject variation without confounding it with intra-subject differences.

To explain the above in the paper we added into the 'Setting and Participants' paragraph in the Methods the following line:

- *In metabolomics studies, analyzing only one sample per subject, especially in large datasets with variable numbers of samples from different subjects, is aimed at ensuring that the data accurately reflects the biological variation of interest, while minimizing biases that can arise from unequal sampling and non-independence of observations.*

Also, it seems more appropriate to collect samples over time in one patient at the end of life to see possible changes in metabolites at the end of life.

This is a very reasonable and logical ask.

We organised 3 different studies over the last 10 years in our region and recruited nearly 400 different people at 8 different sites (4 hospitals and 4 hospices). All of this was done with very little funding. We aimed to collect samples from inpatients 2-3 times a week. This was possible with a small number of patients. For the majority of patients, the collection of repeated samples in the last weeks of life was incredibly difficult to achieve. We were dependent on nurses and health care assistants to remember in their busy schedules to collect a sample or for research nurses to be on site – which may have only been twice a week for half a day. Patients towards the end of life were very frail, for some they may have only urinated once or twice a day and could not supply a sample at a set time. In addition, some patients when they deteriorated were unable to supply a sample unless they were catheterised. This all meant that despite the intention to collect repeated samples over time, this was not always possible for the majority of patients.

We did analyse the samples of patients over time from the few patients we had collected however statistical analyses were consistently underpowered and we have not presented this data. In future research aiming to recruit much larger numbers of patients, we do plan to follow the metabolites of individual patients over time.

We added the following line into the limitations paragraph in the Discussion:

- *Ideally, a time series on a group of individuals each sampled repeatedly over several weeks would give the best models. However, it is very difficult predicting time of death in the last weeks of life, and we don't know who is going to die and when. We did analyse the samples of patients over time from the few patients we had collected however statistical analyses were consistently underpowered and we have not*

presented this data. Therefore, our pragmatic approach was to sample all patients with advanced disease regardless of perceived prognosis and use a cross-sectional approach to identify patterns or predictors of dying.

3. It would be interesting to elaborate on how this would be implemented into future care (when developed into a widely applicable robust clinical tool) as it would be quite expensive and especially time consuming to analyze every patient with LC-QTOF-MS.

You correctly point out that using a LC-QTOF-MS approach for any future test would be expensive and time consuming. We are currently developing our model into a commercial point of care diagnostic test system. Working with a diagnostics research and development company we recently confirmed the technical feasibility of using an enzymatic colorimetric approach to develop a rapid, low-cost commercial assay prototype. Our initial results in both lung cancer and mixed cancer patient cohorts individually and combined was excellent. We have applied for funding to develop this.

We therefore added into the concluding paragraph of the Discussion the following:

- *We are exploring the development of a low cost commercial point of care diagnostic test system.*

Some additional suggestions:

1. Please provide Hazard Ratios with 95%CI when possible

The hazard ratios in Table 2 were developed from the Cox Lasso regression analysis. This is a relatively new algorithm that unfortunately the maths has not caught up with to calculate the confidence intervals. Calculating confidence intervals for hazard ratios in a Cox Lasso regression analysis is challenging due to the nature of the Lasso (Least Absolute Shrinkage and Selection Operator) technique. There are a number of reasons for this:

1. Shrinkage and Selection:

- The Lasso method applies a penalty to the absolute size of the coefficients, which can shrink some coefficients exactly to zero, effectively performing variable selection. This makes the distribution of the estimated coefficients more complex and different from the usual parametric forms assumed in traditional Cox regression.

2. Non-standard Distribution:

- The penalization alters the distribution of the coefficients, making it difficult to apply standard methods for confidence interval estimation. The resulting coefficients do not follow the usual asymptotic normal distribution assumed by maximum likelihood estimation methods.

3. Bootstrap Method:

- One possible approach to estimate confidence intervals in Lasso regression is using bootstrapping, which involves repeatedly resampling the data and refitting the model to create an empirical distribution of the coefficients. However, this is computationally intensive and still may not provide straightforward confidence intervals due to the inherent penalization and selection process.

4. Lack of Analytical Formulas:

- Unlike traditional regression techniques, there are no straightforward analytical formulas for the variance of the coefficients in Lasso regression due to the penalty term. This absence of analytical expressions complicates the derivation of confidence intervals.

5. Regularization Path:

- The Lasso method determines coefficients based on a regularization path, which depends on the tuning parameter (lambda). This dependency adds another layer of complexity in estimating the variability of the coefficients, as different lambda values lead to different sets of selected variables and different levels of shrinkage.

2. Is it known if there is any difference in outcome between different diagnoses, age ranges and smoker status? If so, should the data be adjusted for this difference?

Due to the relatively small sample size it has not been possible to perform these analyses.

3. Some editing is needed on the introduction. For example: the second sentence of the second alinea better connects to the first alinea than the first sentence of the second alinea.

We rewrote the following in the Introduction:

- *Predicting when a patient with advanced cancer is likely to die is a challenge and currently no diagnostic test is available. Globally, there were 19.3 million new cancer cases and almost 10 million cancer deaths in 2020; lung cancer had the highest mortality, responsible for 1.8 million deaths³*

4. Please clarify why has been chosen for lung cancer patients, and why for only lung cancer patients.

When we started this project 10 years ago, we had just performed a 3 month feasibility study in a hospice. We found that patients with lung cancer were the largest cohort of samples collected in the last week of life (five of seven patients). As this was a completely new avenue of research, with no prior studies to guide us, we were concerned that collecting mixed cancer samples could confound our results. Therefore, for our pilot study (this paper) we decided to only recruit lung cancer patients. We are currently seeking funding for a much larger study to recruit patients with mixed cancers.

We included the following line in the 'Patients' paragraph in the Results section:

- *To decrease confounding factors this study only recruited lung cancer patients and not a heterogeneous group of patients with mixture of solid organ cancers.*

Reviewer #3

Major Comments:

Study design

1. Clarity: The manuscript lacks clarity in the study design section. It is difficult to follow how the study was performed.

Your comment was interpreted in 2 different ways

We rewrote the 'Study design' section of the Methods to the following:

- *Urine samples from patients with lung cancer towards the end of life were prospectively collected. Untargeted urine metabolomic studies were conducted using liquid chromatography quadrupole time of flight mass spectrometry (LC-QTOF-MS) to identify metabolites in the urine. To identify the metabolites that change in the last weeks of life, we applied two approaches; ANOVA and volcano plot analysis. To predict the last days of life, a Cox proportional hazards model with a lasso penalty was developed and used. The prediction model was validated on a separate cohort that was collected and analyzed after the initial ("Training") cohort. This validation cohort was also analyzed in a different laboratory using an Orbitrap mass spectrometer. Further KEGG Pathway Analysis was performed, using the significant metabolites identified for the last 2 weeks and last 3 days of life. To identify the pathways affected during the dying process, we then summarised and collated findings from the Volcano plot analysis, ANOVA analyses and KEGG pathway analysis to identify pathways affected during the dying process.*

The second interpretation, was how the data collection was performed:

The following was added to the 'Patients' paragraph in the Results section:

To decrease confounding factors this study only recruited lung cancer patients. TNM staging data were not collected in this study as these are used at cancer diagnosis to predict 5-year survival and guide clinician treatment. We aimed to collect samples from inpatients 2-3 times a week. This was possible with a small number of patients. For the majority of patients, the collection of repeated samples in the last weeks of life was incredibly difficult to achieve. We were dependent on nurses and health care assistants to remember in their busy schedules to collect a sample or for research nurses to be on site – which may have only been twice a week for half a day. Patients towards the end of life were very frail, for some they may have only urinated once or twice a day and could not supply a sample at a set time. In addition, some patients when they deteriorated were unable to supply a sample unless they were catheterised. This all meant that despite the intention to collect repeated samples over time, this was not always possible for the majority of patients. Only 3 patient samples analysed were on some form of chemotherapy (afatinib, pembrolizumab and exemestane) and none of these samples were in the last 6 weeks of life.

2. Study population: Very little information about the study group is presented, with no inclusion or exclusion criteria. Potential confounding factors such as age, sex, therapy (chemo therapy or radiation therapy) and diet are not adequately addressed. These factors could significantly influence the urine metabolome and should be accounted for in the analysis.

For patients with advanced cancer (locally advanced or metastatic), the following were the inclusion and exclusion criteria for our study:

Inclusion criteria:

- Permission to approach patient given by the senior doctor on the clinical team (i.e. Registrar or above) who believes approaching the patient will not cause undue distress.
- Over 18 years of age
- The patient must be able to understand and communicate in English
- Patient is able to provide fully informed consent

Exclusion criteria:

- Patient unable to provide fully informed consent
- Permission not granted to approach patient given by the senior doctor on the clinical team.
- Patients too unwell
- Patients unable to understand English
- Patients likely to be distressed by discussion of the study.

We added the following sentence to the 'Setting and Participants' paragraph in the Methods section:

- *We approached every adult with advanced lung cancer who the treating team thought would not be distressed discussing the study, could understand and communicate in English and had capacity to provide informed consent.*

Regarding diet:

We believe it is an advantage that diet was not controlled for. Our analysis identified 10 food related metabolites that decreased significantly in the last weeks of life. Therefore decreased oral intake was identified as a 'pathway' involved in dying in Table 3. Importantly, our analysis showed a decrease in these metabolites concentration in the last weeks compared to those >3 months from death, that decreased further in the last week of life.

In a separate study (unpublished), we measured serum β -hydroxybutyrate ('ketone') levels in a mixed cancer population. We identified that 'ketones' increased in the last 3 days of life – this would be related to lack of oral intake. There was a small cohort where it increased 3 weeks before death and another small cohort where it did not increase. We believe these may be related to different modes of dying; 'normal', over 2-3 days, 'delayed', over a 7-10 day period and 'rapid', within 12 hours. These modes of dying are observed clinically but not described in the academic literature. Sub-group analysis (with very small numbers) of our Training dataset using Principal Component Analysis was able to separate out these 3 groups. Future work will collect this data and investigate this finding.

We added the following to the eighth paragraph of the Discussion:

- *We believe it is an advantage that diet was not controlled for. Our analysis identified 10 metabolites that decreased significantly in the last weeks of life. Therefore decreased oral intake was identified as a 'pathway' involved in dying in Table 3. Importantly, our analysis showed a decrease in these metabolites concentration in the last weeks compared to those >3 months from death, that decreased further in the last week of life.*

In addition some of these metabolites might be associated with lung cancer and may not be specific to the dying process. This needs to be clarified or refocused to ensure relevance to the study's main objective.

Dying from cancer does not correlate with tumour mass. We report metabolites from the volcano plot analysis that show at least a 2 fold increase (often greater). It is difficult to envision a patient's lung cancer tumour mass therefore doubling or more in the time frames we report. This is not seen clinically.

There are metabolites in our data that are associated with lung cancer. The metabolomics data we obtained from our analyses can be analysed to look at metabolites associated with lung cancer by asking the data different questions. We did get a masters student to do some initial work on lung cancer. We for example were able to identify patients with adenocarcinoma, squamous cell carcinoma and small cell carcinoma. We have not published this work. None of these markers were considered significant in predicting the last weeks of life. Additional work identified novel markers of bone metastases. A preprint for this work is available via BioRxiv (doi: <https://doi.org/10.1101/2024.05.10.593489>).

3. Control Groups: The study appears to lack appropriate control groups. It would be beneficial to compare this new method with current methods and include control samples from healthy individuals. This would help in establishing the specificity and relevance of the findings.

For our study, a traditional control group was not appropriate. This study was not a comparison to healthy controls – it was a study of people with advanced cancer and observing patterns and predictors of death. Predictive modelling relies on identifying patterns within the data that directly relate to the outcome of interest i.e. death. The use of techniques like cross-validation (our validation cohort) ensures the model is robust and can generalize well to new data, eliminating the need for a traditional control group.

Volcano plot analysis eg for the last 4 weeks, 2 weeks, 5 days and 3 days were compared to all samples greater than 4 weeks, 2 weeks, 5 days and 3 days respectively. We made no attempt to 'optimise' our data by comparing to a different group e.g. samples >3 months from death.

For ANOVA analysis, we grouped the samples into the last week of life, the second last week, the third last week, weeks 4-11 and weeks 12 plus. The analysis of these groupings yielded several significant results. Therefore the groupings we used were able to detect changes in metabolites in

the last weeks of life and both specific and relevant to the aim of our research. We are uncertain what a grouping of age matched controls without cancer would have added.

We added the following to the 'Prediction model - Validation cohort' section in the Results:

Predictive modelling relies on identifying patterns within the data that directly relate to the outcome of interest i.e. death. The use of techniques like cross-validation (our validation cohort) ensures the model is robust and can generalize well to new data, eliminating the need for a traditional control group.

4. Time Points: The study seems to include four time points when refereeing to Table 3

We clarify queries re Table 3 when we address comment 10.

however according to line 482 "In order not to bias the results, only one sample per patient, the one closest to death, was included for analysis." Please clarify. In addition, the study seems to have very few data points which may be insufficient for capturing the full dynamics of the dying process. Using more time points could provide more comprehensive insights, and the reviewer doesn't understand the risk of bias?

In metabolomics studies, analyzing only one sample per subject, especially in large datasets with variable numbers of samples from different subjects like ours, can help in reducing bias and ensure the reliability of the results. Here are several reasons why this approach is often used:

1. **Avoiding Pseudoreplication:** When multiple samples from the same subject are treated as independent data points, it can lead to pseudoreplication. Pseudoreplication artificially inflates the sample size and can result in misleading statistical conclusions. Analyzing only one sample per subject helps in treating each subject as a single data point, thus avoiding this issue.
2. **Reducing Intra-Subject Variability:** Multiple samples from the same subject can exhibit variability due to biological or technical factors. By choosing a single representative sample per subject, the analysis can focus on inter-subject variability rather than intra-subject variability, which is often the main interest in biological studies.
3. **Simplifying Statistical Analysis:** Statistical methods used in metabolomics, such as multivariate analysis, are more straightforward and interpretable when each subject contributes a single data point. This simplifies the modeling and interpretation of the results.
4. **Balancing Sample Size:** In studies with an unequal number of samples per subject, analyzing only one sample per subject ensures a balanced dataset. This balance is crucial for statistical methods that assume equal representation of subjects, thereby reducing potential bias introduced by over-representation of subjects with more samples.
5. **Ensuring Independence:** Analyzing multiple samples from the same subject can violate the assumption of independence, which is a key requirement for many statistical tests. Using one sample per subject helps maintain the independence of observations, leading to more robust and valid statistical inferences.
6. **Focus on Biological Variation:** The primary goal of many metabolomic studies is to understand biological variation between subjects, such as differences due to disease states, treatments, or other factors. By analyzing one sample per subject, researchers can more accurately capture this inter-subject variation without confounding it with intra-subject differences.

Ultimately, the strategy of analyzing one sample per subject is aimed at ensuring that the data accurately reflects the biological variation of interest, while minimizing biases that can arise from unequal sampling and non-independence of observations.

To explain the above we added in the following line into the 'Setting and Participants' paragraph of the Methods:

- *In metabolomics studies, analyzing only one sample per subject, especially in large datasets with variable numbers of samples from different subjects, is aimed at ensuring that the data accurately reflects the biological variation of interest, while minimizing biases that can arise from unequal sampling and non-independence of observations.*

We also added the following into the limitations paragraph of the Discussion:

- *Ideally, a time series on a group of individuals each sampled repeatedly over several weeks would give the best models. However, it is very difficult predicting time of death in the last weeks of life, and we don't know who is going to die and when. We did analyse the samples of patients over time from the few patients we had collected however statistical analyses were consistently underpowered and we have not presented this data. Therefore, our pragmatic approach was to sample all patients with advanced disease regardless of perceived prognosis and use a cross-sectional approach to identify patterns or predictors of dying.*

Interpretation

5. Interpretation of Results: The interpretation of the results, particularly regarding false positives, i.e. when the prediction was incorrect, is not clear. The manuscript should discuss why certain individuals lived longer or shorter than predicted based on the metabolome data.

In the paper we did write the following:

- *The model calibration was stable at 30, 20 and 10 days; slightly overestimating those likely to die with a High-risk score and underestimating dying with a Low-risk score.*

This is an interesting question. This paper is the first description of our model that predicts dying. As you are also aware, it is the first paper to perform an -omics study in patients in the last days and weeks of life. The exploratory nature of our work is both important and interesting. Since it is exploratory – many important questions are being raised. We therefore cannot discuss why certain individuals lived longer or shorter than predicted, yet. The scientific literature to answer this is lacking. We plan to address this in future research.

However, to discuss your point. As a palliative care clinician responding to your question. Despite all the advances of the 21st century we do not know how most people die from cancer (or most other diseases). There are acute causes of death but these are not what the majority of people die from (in my experience). This was reviewed in the June 2024 nature reviews paper

(<https://www.nature.com/articles/s41568-024-00708-4>). Therefore we do not expect our model to detect all (eg acute) dying processes from cancer. A process for dying has been proposed but never proven. What has never been proposed – although observed by every clinician who works in a hospice, it that different people can die in different ways; ‘normal’ over 2-3 days, ‘rapid’ in less than 12 hours and ‘delayed’ when people are on the cusp of death for a week or so. In future work we will collect this data when we collect samples and explore this.

We also are exploring converting our model into a point of care assay. We are working with a diagnostics research and development company on this. We have technical proof of concept to use a colorimetric enzymatic approach. In the course of getting this data we observed that our test was highly specific (nearly 95%) but not as sensitive (approx. 70%) in a small mixed cancer population. The test does not have many false positive, it has a number of false negatives – meaning that it is not picking up a cohort of patients who are dying – this we believe, is likely related to different modes of dying. Future work will explore this cohort.

6. Circadian Rhythm and Analyte Measurement: The manuscript does not discuss whether the analytes exhibit circadian rhythms that might be influenced by factors such as diet. It is important to specify when the measurements were taken to ensure consistency and accuracy.

Due the variability of collection of samples throughout the day there was no opportunity to ensure urine samples were collected at the same time. In addition, it is difficult to control for circadian rhythm – not only do different people have different rhythms (different chronotypes), patients with advanced cancer can have a disrupted circadian rhythm.

Interestingly, cortisol and tryptophan metabolism (both controlled by the circadian rhythm) both increased in the last days of life. The increase in tryptophan would not have been due to food intake as patients in the last days of life have decreased oral intake. In addition, tryptophan metabolism is thought to control the circadian rhythm. This raises the interesting question – is the circadian rhythm in people who are dying disrupted?

We added the following line to the 7th paragraph of the discussion:

- *Both cortisol and tryptophan metabolism are influenced by the circadian rhythm yet in the last days of life both cortisol production and tryptophan metabolism increase.*

Method

7. Calibration and Batch Effects: The manuscript mentions calibration points but does not clearly explain how they were used. Furthermore, it is stated that samples were run in batches, but it is unclear how batch effects were corrected. More details on the calibration process and batch effect correction are needed to assess the reliability of the results.

We took the decision not to apply batch correction. Our inclusion of the same pooled QC group samples across each batch enabled assessment of batch effects – these effects were considered minimal for several reasons.

a) the close overlay of TICs of replicate injections of the same QCs within and across batches 1 & 2.

b) metabolites considered in the analysis all were observed in 100% of QC pools from at least one sample group and had CV <25% in data from replicate injections of each QC group across batches 1 and 2 as part of our QC workflow. The authors feel this justifies our chosen approach given the metabolomics field is yet to reach consensus on batch correction procedures in LC/MS datasets

c) Samples were assigned random numbers (Microsoft Excel) and sorted in order of these random numbers. This meant that sample allocation to batch 1 or 2 and run order was randomised.

The following is now included in the 'Urine collection, storage and preparation' paragraph in the Methods:

- *Pooled quality control samples were created following the protocol described by Norman et al.²⁷. For each sample group (time before death), a separate representative pool was created by pooling an equal volume of each individual urine sample for quality control purposes. An overall pool was also created by pooling equal proportions of the above group pools.*

Re calibration.

Calibration of the instrument (e.g. accurate mass calibration) was performed according to manufacturer guidelines.

If calibration refers to our feature finding approach - our workflow is fully described in the Norman et al. (2019) paper published in Clinical Chemistry. It is the laboratory practice to do a mass calibration before each run.

8. Group Creation: The criteria for creating the three groups mentioned in the study are not described. Clear definitions and justifications for group divisions are necessary for understanding the comparative analyses.

Kaplan-Meier curves are often used to compare survival probabilities between different groups (e.g., treatment vs. control). When assessing the calibration of our model, we split the patients into three groups based on their predicted risk of death within 30 days. The predicted and observed deaths were compared to assess model calibration. As a further visual tool, we plotted Kaplan-Meier curves for these three risk groups to demonstrate that the groups do indeed have very different risk of death.

We included the following line in the 'Prediction model - Training cohort' paragraph in the Results section of the paper:

- *Stratifying subjects into risk groups based on model predictions and plotting their survival curves can reveal a model's discriminatory power. We therefore divided our patients up into three equally sized groups based on their risk of death in 30 days. Kaplan-Meier survival curves were plotted for patients classified as Low, Medium*

and High risk of dying (see Figure 1 A, log-rank test $p < 0.001$). This approach allows for the visualization and comparison of survival probabilities among individuals grouped into different risk categories based on their predicted likelihood of death within the last 30 days of life, as estimated by the Cox Lasso regression model. The Kaplan-Meier survival curves show it is possible to assign individuals into different risk categories *i.e. the differing levels of predicted risk were reflected in the numbers of observed deaths in each group. The group of patients with the highest predicted risk of death, did indeed have the highest numbers of deaths. Conversely, the group with lowest predicted probabilities experienced the fewest deaths.*

9. Figure 2: There appears to be a discrepancy in Figure 2, with two metabolites missing. This needs to be addressed and corrected.

Indeed the figure only shows 5 metabolites. Cox Lasso Regression analysis identified seven metabolites however two of these metabolites were not identified by ANOVA analysis as being significant. The ANOVA looks at whether values of the same biomarker change over time. Whereas the Cox model looks at whether a value of a biomarker can predict how long until death. These are different approaches, and it doesn't necessarily follow that the same metabolites would be identified by both approaches. Thus there are 5 metabolites in the figure.

In the paper we added the following to the second paragraph of the 'Prediction model – Training cohort' section in the Results:

- *While Cox Lasso Regression analysis identified seven metabolites, five of these metabolites were identified by ANOVA analysis as being significant.*

And we rewrote the title to Figure 2 to the following:

- *Plots for each metabolite from the prediction model identified as significant by ANOVA over the last 12 weeks of life versus more than 3 months from death.*

Figure 2 suggests that all metabolites decrease, which could be due to dietary differences and reduced appetite. This possibility should be explored and discussed.

Reduced oral intake is indeed very important. I can see why you think the metabolites decrease. They in fact increase. The last week of life is on the left hand side of the figure.

Two metabolites in the prediction model decrease; gluconic acid which is abundant in food and Histidinyl-hydroxyproline, a bone metabolite. As regards dietary differences, in Table 3 we listed decreased oral intake as a 'pathway' and listed the following metabolites identified to have decreased significantly:

Caffeic Acid ↓

Caffeine ↓
Ferulic acid ↓
Paraxanthine ↓
Quinic acid ↓
Rosmarinic acid ↓
Tartaric acid ↓
Theobromine ↓
Theophylline ↓
Trigonelline ↓

Also in the discussion when we discuss the model there is the following sentence:

“Gluconic acid, abundant in plants, decreases and suggests decreased oral intake.”

In addition we added the following to the 8th paragraph in the Discussion:

- *We believe it is an advantage that diet was not controlled for. Our analysis identified 10 metabolites that decreased significantly in the last weeks of life. Therefore decreased oral intake was identified as a ‘pathway’ involved in dying in Table 3. Importantly, our analysis showed a decrease in these metabolites concentration in the last weeks compared to those >3 months from death, that decreased further in the last week of life.*

10. Table 3 Metabolites: It is unclear which metabolites Table 3 is based on. The manuscript should specify whether it includes all metabolites and all participants or only a subset for example the test set.

The following is now included under the title of Table 3.

- *This table is a summary of all the metabolites identified as changed significantly from the volcano plot analysis and ANOVA analysis of the Training dataset. It includes a summary of the KEGG Pathway analysis for the last 2 weeks and last 3 days.*

Additionally, information on how the analytical quality of the data was ensured is required.

Data from the same pooled QC samples interspersed throughout each analytical run were central to our quality assurance procedure.

The following is now in the ‘Urine collection, storage and preparation’ paragraph in the Methods:

- *Pooled quality control samples were created following the protocol described by Norman et al.²⁷. For each sample group (time before death), a separate representative pool was created by pooling an equal volume of each individual urine sample for quality control purposes. An overall pool was also created by pooling equal proportions of the above group pools.*

11. Exclusion of Medication Metabolites: The manuscript states that medications and related metabolites were excluded. It is important to consider that some of these unidentified metabolites could be drug artifacts. A more thorough explanation of the exclusion criteria and potential implications is needed.

For every sample collected we had a list of medications a patient was taking. None of the metabolites of these medications were detected – this is despite interrogating all of the major spectral libraries in metabolomics research, suggesting these metabolites have not been well-characterized before. In addition, the unknown metabolites were present in almost all samples suggesting the drug artifact possibility extremely unlikely.

We included the following in the ‘Setting and Participants’ paragraph of the Methods:

- *The list of medications a patient was on for the previous 24 hours was recorded for each sample.*

To discuss your comment further, we added the following to 9th paragraph in the Discussion:

- *For those samples, predominantly collected in months before death, patients were admitted to hospital. The main reason a patient with advanced cancer was admitted to hospital was for an infection and therefore these patients would have been on IV antibiotics – predominantly Tazocin. For those patients in the last days of life, the majority are not on antibiotics. Therefore, an increase in antibiotics metabolites in the last days of life would not be detected. In the last days of life, patients normal medications would have stopped due to decreased oral intake or stopped for clinical reasons. Therefore, the metabolites of non-symptom related medications (e.g. antihypertensives etc) would be expected to decrease. However, certain symptoms can increase towards the end of life. To manage these, medications would either be added or possibly increase. In general there is a limited number of medications used to control symptoms at the end of life e.g. morphine for pain, midazolam for terminal agitation, glycopyrronium for respiratory secretions. The well characterised metabolites of these medications were not detected in our analyses.*

12. While the study is described as prospective, it is not clear whether all samples were analyzed at the same period after collection. This is further complicated by having two “separate” method sections with differences in amount of information.

The training set of samples was collected from 2016-2019 and analysed in 2019. The validation set of samples was collected 2019-2020. These were analysed in 2022.

We added the following to the ‘Prediction model - Validation cohort’ section in the Results:

- *The Validation cohort was recruited after the initial Training cohort and analyzed 3 years after the Training cohort.*

Minor Comments:

1. Sample Preparation and Storage: There is insufficient information regarding the preparation and storage of samples before analysis. Detailed descriptions of these processes are essential for reproducibility and understanding potential sources of variability.

We added the following to the paper:

➤ *Urine collection, storage and preparation*

Random urine samples were collected twice per week from participants into universal (20 mL) and glass (5mL) containers. An anonymised record of medication administered in the previous 24 h was recorded. Samples were stored on site at -20°C before subsequently being transferred to the University of Liverpool for further storage at -80°C. Individual patient samples were thawed at room temperature, vortexed and supernatants separated into four replicate aliquots in individual 96-well plates (Waters, UK) which were stored at -80 °C until analysis by one of four different methods; two different chromatography conditions in negative and positive ionisation polarity. Pooled quality control samples were created following the protocol described by Norman et al. ²⁷ . For each sample group (time before death), a separate representative pool was created by pooling an equal volume of each individual urine sample for quality control purposes. An overall pool was also created by pooling equal proportions of the above group pools. Analysis of individual and pooled samples was performed following dilution of 1:3 urine:deionised water (DIRECT-Q 3UV Millipore water purification system) as previously described by Norman et al. ²⁷ .

2. HMDB Database: Consider using the HMDB database for metabolite identification, as it is a well-established resource that could enhance the study's reliability and comparability.

We did search HMDB. HMDB is one of the databases included in the SIRIUS package.

We added the following (in bold) to the 'Prediction model – Training cohort' in the Results section:

- *Compound identification of unknown metabolites in our model was attempted using public spectral libraries including SIRIUS (**which included HMDB database**), Metlin, Mona and mzCloud.*

AND in the following (in bold) to the 'Feature extraction' paragraph in the Methods:

- *Compound identification of unknown metabolites in our model was attempted by matching of our acquired MS/MS spectra against the spectral libraries METLIN ²⁹*

*(Agilent METLIN metabolite PCDL accurate mass library (build 07.00)), METLIN online³⁰, MoNA³¹ and also using SIRIUS software (version 5.6.3) tools including **the HMDB database**, the SIRIUS molecular formula identification and CSI:FingerID fingerprint prediction.*

3. Abbreviations: The manuscript uses abbreviations such as LC-QTOF and LC-Orbitrap. Both instruments are MS/MS systems, but the abbreviation MS/MS is often associated with triple quad instruments rather than high-resolution instruments like Orbitrap. It is recommended not to abbreviate Orbitrap with UHPLC-MS/MS, as this can be misleading.

We deleted the abbreviation.

REVIEWERS' COMMENTS:

Reviewer #1 (Remarks to the Author):

The authors carefully revised the manuscript and properly addressed the reviewers' comments.

This manuscript is interesting and will provide valuable insights not only for patients but also for doctors in making medical decisions.

This manuscript can be considered for publication.

Minor comment: The letter resolution in Figure 2 is not clear, particularly for "week 01 ~ week 12+".

Reviewer #2 (Remarks to the Author):

Thanks for very carefully reflecting on the comments of all reviewers

Reviewer #3 (Remarks to the Author):

I am very pleased with the revisions made to the manuscript and appreciate the thorough responses to the reviewer comments. The improvements have enhanced the quality of the work, and I now believe it is ready for publication.

Response to Reviewers' comments

August 2024

Reviewer #1

The authors carefully revised the manuscript and properly addressed the reviewers' comments.

This manuscript is interesting and will provide valuable insights not only for patients but also for doctors in making medical decisions.

This manuscript can be considered for publication.

Minor comment: The letter resolution in Figure 2 is not clear, particularly for "week 01 ~ week 12+".

Comments:

We have renamed Figure 2 to Figure 3 and increased the font size on the x-axis

Reviewer #2

Thanks for very carefully reflecting on the comments of all reviewers

Comments:

Thankfully nothing to address!

Reviewer #3

I am very pleased with the revisions made to the manuscript and appreciate the thorough responses to the reviewer comments. The improvements have enhanced the quality of the work, and I now believe it is ready for publication.

Comments:

Also thankfully nothing to address!